# Stability of peatland carbon to rising temperatures

R.M. Wilson [1,*], A.M. Hopple[2,*], M.M. Tfaily[3], S.D. Sebestyen[4], C.W. Schadt[5], L. Pfeifer-Meister[2], C. Medvedeff[6], K.J. McFarlane[7], J.E. Kostka[8], M. Kolton[8], R.K. Kolka[4], L.A. Kluber[5], J.K. Keller[6], T.P. Guilderson[7], N.A. Griffiths[5], J. P. Chanton[1], S.D. Bridgham[2] & P.J. Hanson[5]

Peatlands contain one-third of soil carbon (C), mostly buried in deep, saturated anoxic zones (catotelm). The response of catotelm C to climate forcing is uncertain, because prior experiments have focused on surface warming. We show that deep peat heating of a 2 m-thick peat column results in an exponential increase in $CH_4$ emissions. However, this response is due solely to surface processes and not degradation of catotelm peat. Incubations show that only the top 20–30 cm of peat from experimental plots have higher $CH_4$ production rates at elevated temperatures. Radiocarbon analyses demonstrate that $CH_4$ and $CO_2$ are produced primarily from decomposition of surface-derived modern photosynthate, not catotelm C. There are no differences in microbial abundances, dissolved organic matter concentrations or degradative enzyme activities among treatments. These results suggest that although surface peat will respond to increasing temperature, the large reservoir of catotelm C is stable under current anoxic conditions.

[1] Earth, Ocean and Atmospheric Sciences, Florida State University, 117 N Woodward Avenue, Tallahassee, Florida 32306, USA. [2] Institute of Ecology and Evolution, University of Oregon, Eugene, Oregon 97403, USA. [3] Environmental Molecular Sciences Laboratory—Pacific Northwest National Laboratory, Richland, Washington 99354, USA. [4] USDA Forest Service Northern Research Station, Grand Rapids, Minnesota 55744, USA. [5] Oak Ridge National Laboratory, Oak Ridge, Tennessee 37831, USA. [6] Schmid College of Science and Technology, Chapman University, Orange, California 92866, USA. [7] Lawrence Livermore National Laboratory, Livermore, California 94550, USA. [8] School of Biological Sciences and School of Earth and Atmospheric Sciences, Georgia Institute of Technology, Atlanta, Georgia 30332, USA. * These authors contributed equally to this work. Correspondence and requests for materials should be addressed to R.M.W. (email: rmwilson@fsu.edu).

Peatlands store a globally significant fraction of the world's carbon (C) in deep recalcitrant peat[1], which, if destabilized, could result in catastrophic positive feedbacks to climate warming. However, all soil-warming experiments exploring the response of peatland C banks to climate forcing to date have used limited surface-warming techniques (generally $+1\,°C$), ignoring the effects on deeper buried peat layers[2,3]. Thus, despite the established significance of peatlands in the global C cycle, their response to future climate change remains poorly constrained[4,5], because under long-term warming, deep soil temperatures will increase in parallel with atmospheric temperatures[6–9]. The large reservoirs of C at depth (well over a metre) mean that this largely ignored C fraction could play a significant—although as yet unquantified—role in future climate change.

To address this gap, the Spruce and Peatland Responses Under Climatic and Environmental Change (SPRUCE; http://mnspruce.ornl.gov) experiment is assessing how northern peatland ecosystems react to a changing climate with a regression-based, ecosystem-scale climate manipulation that incorporates deep peat heating (DPH) from $+0$ to $+9\,°C$ above ambient to a depth of 2 m (ref. 10). The SPRUCE experiment is located at the S1 bog within the Marcell Experimental Forest (Minnesota, USA)[11]. Ultimately, the SPRUCE experiment will include both above- and below-ground warming, as well as ambient and elevated air $CO_2$ concentrations in a multifactorial experimental design. However, below-ground DPH was initiated first and is the sole treatment reported here. Deep peat is expected to warm naturally, in parallel with surface warming, due to the propagation of heat downwards into the peat column. However, to achieve this effect in a tractable spatial scale for experimentation, active heating of the peat at depth is required[7]. Although the highest climate trajectories project temperature increases up to $+8.3\,°C$ ($\pm 1.9\,°C$) in the Arctic between 2081 and 2100 (ref. 12), the $+9\,°C$ treatment employed in this study is an upper limit on what can be expected under the most extreme scenarios. This treatment design allows for the exploration of

nonlinear and threshold response surfaces to temperature change (for example, see Eppinga et al.[13]). SPRUCE represents a novel experiment that provides the first field-scale examination of the response of deep C and associated heterotrophic microbial communities to warming.

Herein we find that $CH_4$ emissions increase exponentially with deep heating of a peatland; however, this response is due solely to the warming effect on surface peat ($<30\,cm$ deep). Peat from deeper depths does not respond to elevated heating treatments. Radiocarbon analyses suggest that $CO_2$ and $CH_4$ emissions result from decomposition of surface-derived dissolved organic C (DOC) rather than from degradation of ancient catotelm peat. Microbial results confirm these findings and enzyme activities suggest that oxidative activity may be confined to shallower peat as well. Cumulatively, these multiple lines of evidence suggest that future climate warming will stimulate higher $CH_4$ emissions due to increased production in surface peat, but that deep catotelm C will remain stable under elevated temperatures.

## Results

**Physical responses.** From June 2014 through August 2015, DPH treatments to $>2\,m$ depth were established within ten 12 m-diameter plots (0, $+2.25$, $+4.5$, $+6.75$ and $+9\,°C$, relative to ambient, in duplicate, see Fig. 1 for schematic of the site) within the S1 bog, following the approach described in Hanson et al.[6]. Target temperature differentials were achieved at 2 m depth by September 2014 (Fig. 2a). Temperature differentials among plots were greatest below 50 cm and diminished towards the surface as the result of heat loss (Fig. 2b).

During the DPH experiment, we measured water table depth in each plot (30 min measurement frequency) and did not observe—nor did we expect—any changes in water table elevation that was attributable to the deep peat warming treatment. Water tables were usually within 20 cm of the mean hollow surface and fluctuated over a $\sim 40\,cm$ range due to rainfall, snowmelt inputs,

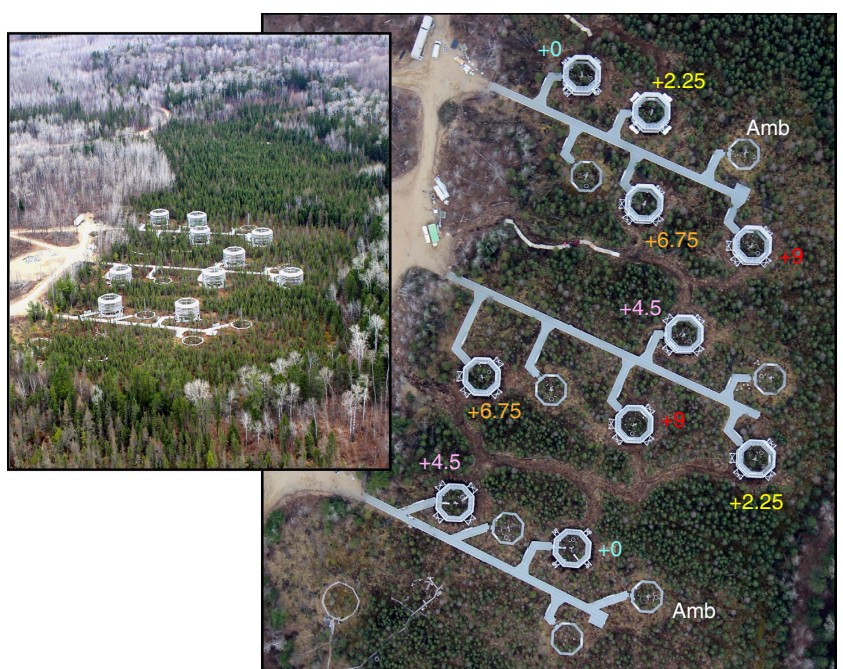

**Figure 1 | Schematic of the SPRUCE site located in northern Minnesota.** Three boardwalks transect the site, with experimental treatments plots branching radially off of those boardwalks. Numbers indicate the target temperatures, relative to ambient conditions, established within each enclosure. 'Amb' plots indicate that no temperature treatment has been added. Inset shows an aerial overview of the site with the experimental enclosures installed in the context of the surrounding bog.

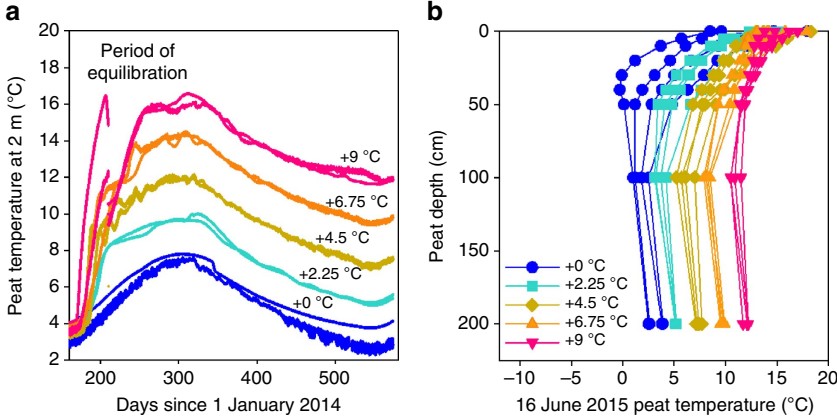

**Figure 2 | Peat temperatures throughout DPH treatment period.** The seasonal progress of (**a**) absolute peat temperatures at 2 m below the hollow surface throughout the DPH treatment period and (**b**) the temperature depth profiles associated with the 16 June 2015 coring event. This coring event took place 10 months after the deep peat temperature differentials were stable. In **a**, blue denotes control ($+0\,°C$) plots, green denotes $+2.25\,°C$ plots, gold denotes $+4.5\,°C$ plots, orange denotes $+6.75\,°C$ plots, and red denotes $+9\,°C$ plots. In **b**, circles denote control ($+0\,°C$) plots, squares denote $+2.25\,°C$ plots, diamonds denote $+4.5\,°C$ plots, triangles denote $+6.75\,°C$ plots and inverted triangles denote $+9\,°C$ plots. In the absence of air warming during this phase of the experiment, anticipated energy loss reduced the separation among treatment temperatures at the surface.

near-surface lateral flow and evapotranspiration, but with no apparent effect of DPH. In addition, water table dynamics inside experimental plots mirrored those measured in ambient reference plots and the surrounding bog.

**$CH_4$ and $CO_2$ fluxes**. $CH_4$ and $CO_2$ fluxes showed seasonal effects with the highest fluxes observed during the peak growing season (June/July; Fig. 3 and Supplementary Fig. 1). $CH_4$ flux increased exponentially with deep soil warming during the ice- and snow-free season (Fig. 3a,c). However, $CH_4$ flux was dramatically reduced by snow and ice cover during the winter and no warming effect was observed (Fig. 3b). Dark $CO_2$ flux was not correlated with deep soil temperature during any measurement time (Supplementary Fig. 1).

**Incubation results**. In laboratory incubations of peat within 1 °C of in situ temperatures, surface peat (20–30 cm below the hollow surface) had greater $CH_4$ and $CO_2$ production rates than peat from deeper depths, except at the coldest temperatures (General Linear Model, $P < 0.001$). $CH_4$ and $CO_2$ production in surface peat significantly increased with temperature ($P < 0.001$, Fig. 4a and Supplementary Fig. 2a). In contrast, no relationship between temperature and $CH_4$ or $CO_2$ production was observed in incubations of peat from depths deeper than 25 cm ($P \geq 0.97$, Fig. 4b and Supplementary Fig. 2b). The $CO_2:CH_4$ ratio was negatively correlated with temperature in surface peat ($P \leq 0.001$, Fig. 5), but not in deeper peat ($P > 0.1$).

Replicate anaerobic incubations from each treatment/depth were conducted at a common temperature (20 °C), to determine whether any legacy effects persisted after removing the direct effects of temperature following 13 months of DPH. In agreement with the incubations at in situ temperatures, we observed greater $CH_4$ and $CO_2$ production in surface peat relative to that of deeper peat in the legacy effect incubations. Although $CH_4$ and $CO_2$ production rates were higher at 20 °C than in the incubations at in situ temperatures, there was no correlation between production rates and the initial treatment temperature (Supplementary Fig. 3).

**Field geochemistry results**. Comparing the radiocarbon ($^{14}$C) content of the dissolved $CO_2$ and $CH_4$ with that in the DOC and peat allowed us to differentiate the source of organic matter fueling heterotrophic respiration—either recent photosynthate (DOC) or ancient catotelm peat[14,15]. In all plots and depths, the $\Delta^{14}$C data indicate that $CH_4$ and dissolved inorganic C (DIC) were relatively young, $^{14}$C-enriched relative to the peat and indistinguishable from the $\Delta^{14}$C of the DOC (Fig. 6a). The $\alpha_C$, which is a measure of the difference between $\delta^{13}CO_2$ and $\delta^{13}CH_4$, increased with depth in all treatment plots, consistent with a shift from acetoclasty at shallow depths ($<50$ cm) to hydrogenotrophy at deeper depths (Fig. 6b), although no effect of temperature on $\alpha_C$ was observed. DIC and dissolved $CH_4$ concentrations were also stable across all treatments (not shown) as were total DOC concentrations (Supplementary Fig. 4).

**Microbial results**. Based on 12 million gene sequences retrieved from 220 samples, microbial community composition and diversity were similar across all temperature treatments and between pre- and post-DPH measurements (Fig. 7 and Supplementary Fig. 5). Both C decomposition and microbial community structure exhibited strong vertical stratification, similar to pre-treatment findings (for example, see ref. 16). The majority of microbial populations ($\sim 70\%$) were taxonomically affiliated with *Proteobacteria* and *Acidobacteria* (Supplementary Fig. 6), and microbial diversity decreased with depth (not shown). Members of the *Alphaproteobacteria* and *Acidobacteria* classes decreased in relative abundance with depth, whereas *Deltaproteobacteria* and TM1 increased in the catotelm (Supplementary Figs 7 and 8). One year after treatment initiation, quantitative PCR of the 16S and 18S ribosomal RNA genes also showed decreasing overall bacterial and fungal abundance with depth but no significant response to temperature (Supplementary Fig. 9). Similarly, Archaeal abundances and *methyl coenzyme A reductase* (*mcrA*) gene abundance did not show a temperature treatment effect (Supplementary Fig. 9d) nor did the relative abundance of methanogens out of total Archaea (Supplementary Fig. 10). Enzyme activity potentials were consistent across temperature treatments after 13 months of heating (Supplementary Fig. 11 and 12). Although the highest phenol oxidase activity occurred in shallow peat (0–30 cm; Supplementary Fig. 11), the highest phenol peroxidase activity occurred in the catotelm ($>30$ cm; Supplementary Fig. 12).

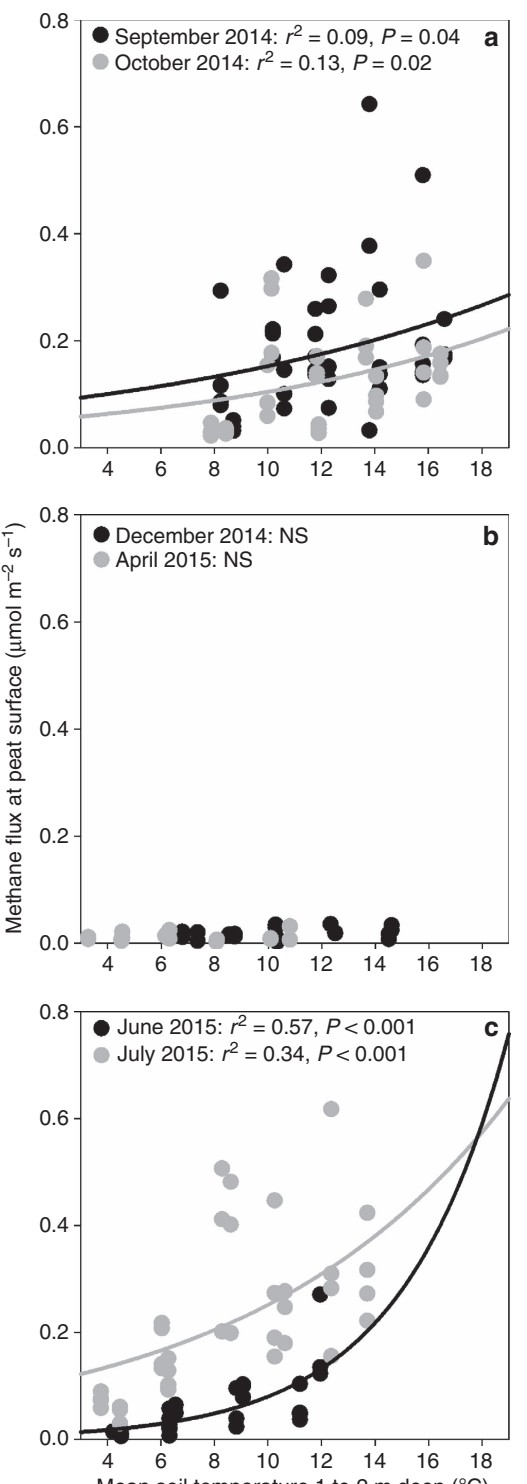

**Figure 3 | Seasonal CH$_4$ fluxes from S1 bog.** Seasonal CH$_4$ flux vs *in situ* temperatures from 1.2 m diameter collars during (**a**) fall 2014, (**b**) winter 2014-2015 and (**c**) summer 2015. Black and grey dots distinguish between daily averages for two different sampling times during each season. Significant correlations between flux and temperature as exponential regressions are indicated on the graphs by black lines for September (**a**) and June (**c**), and grey lines for October (**a**) and July (**c**).

## Discussion

DPH up to 9 °C above ambient failed to stimulate catotelm C decomposition in this ombrotrophic bog within the first 13 months of the DPH experiment. The absence of air warming

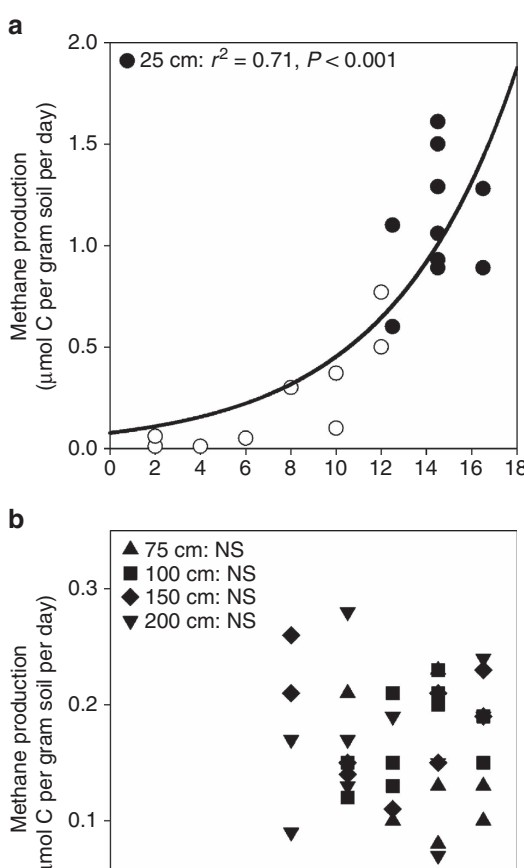

**Figure 4 | CH$_4$ production in anaerobic incubations.** Temperature response of CH$_4$ production from (**a**) surface and (**b**) deep peat samples that were anaerobically incubated within 1 °C of *in situ* temperatures after ∼4 (closed symbols, September 2014) and 13 (open symbols, June 2015) months of deep peat warming. Circles represent values from 25 cm, triangles represent results from 75 cm, squares represent results from 100 cm, diamonds represent results from 150 cm and inverted triangles represent results from 200 cm. Temperatures reflected *in situ* temperatures at time of collection. The temperature response of deep peat (**b**) for each season was analysed separately due to a distinct bimodal distribution. Dark line indicates significant regression results. NS, not significant.

during this phase of the experiment resulted in heat loss at the surface, creating less temperature separation in shallow peat relative to peat below 50 cm (Fig. 2b). Once the experimental plots reached target temperature differentials, CH$_4$ flux but not the net ecosystem respiration—as measured by dark CO$_2$ flux— increased exponentially with deep soil temperature (Fig. 3) despite reduced warming at the surface due to energy losses. The temperature response was maximal during the peak growing season (Fig. 3c) and CH$_4$ flux was dramatically reduced by snow and ice cover during the winter (Fig. 3b).

Consistent with these field emission results, when peat was incubated anaerobically within 1 °C of *in situ* temperatures, CH$_4$ and CO$_2$ production in surface peat (20–30 cm below the hollow surface) increased with temperature (Fig. 4a and Supplementary Fig. 2a, respectively). This layer is within the acrotelm[17], but was consistently anaerobic at the time of sampling. Importantly, no relationship between temperature and CH$_4$ or CO$_2$ production

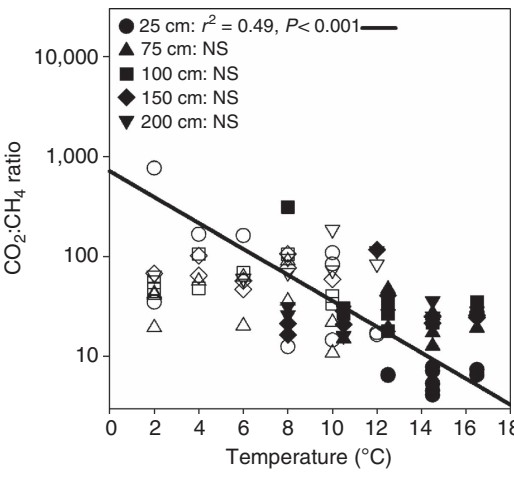

**Figure 5 | CO$_2$:CH$_4$ ratios in anaerobic incubations.** Peat samples were collected from five depths and anaerobically incubated within 1 °C of *in situ* temperatures after ~4 (closed symbols; September 2014) and 13 (open symbols; June 2015) months of deep peat warming. The circles indicate results from peat collected from 25 cm, triangles indicate results from 75 cm, squares indicate results from 100 cm, diamonds indicate results from 150 cm and inverted triangles represent results from 200 cm. The line indicates the regression result for the 25 cm incubations. NS, not significant. The log scale is worth noting.

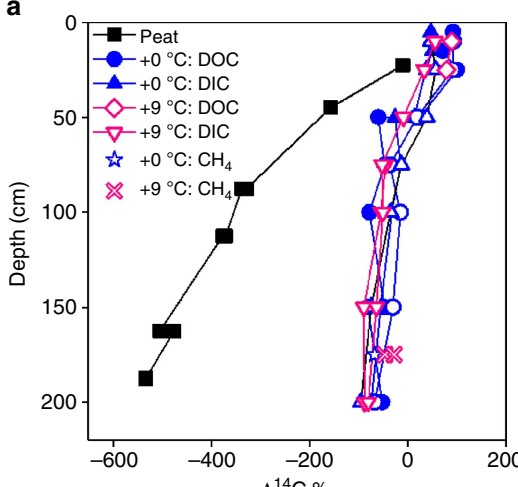

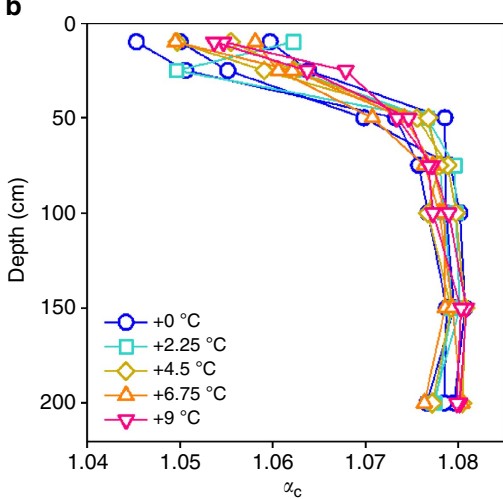

**Figure 6 | Isotopic composition of respiration products and substrates.** Depth profiles of (**a**) $^{14}$C for solid peat, DOC, CH$_4$ and dissolved CO$_2$ (DIC), and (**b**) differences in stable carbon isotopic ($\delta^{13}$C) composition between DIC and CH$_4$ ($\alpha_C = [(\delta^{13}CO_2 + 1000)/(\delta^{13}CH_4 + 1000)]$) with depth during DPH (June 2015). In **a**, closed symbols represent values from control plots prior to DPH when no treatment (that is, +0 °C) was applied, open symbols represent values from +9 °C treatment plots during DPH (June 2015). In **a**, black squares denote radiocarbon peat values, closed blue circles denote radiocarbon DOC values from control (0 °C) plots, blue triangles denote radiocarbon DIC values from control plots, magenta diamonds denote radiocarbon DOC values from +9 °C plots, magenta inverted triangles denote radiocarbon DIC values from +9 °C plots, blue stars denote radiocarbon CH$_4$ values from control plots, and magenta x's denote radiocarbon CH$_4$ values from +9 °C plots. In **b**, blue circles denote $\alpha_C$ from control plots, turquoise squares denote $\alpha_C$ from +2.25 °C plots, gold diamonds denote $\alpha_C$ from +4.5 °C treatment plots, orange triangles denote $\alpha_C$ from +6.75 °C plots and magenta inverted triangles denote $\alpha_C$ from +9 °C plots. Note the age difference between solid peat and all DOC and DIC values.

was observed in incubations of peat from deeper depths, implying that the increased CH$_4$ emissions observed in the field were largely driven by surface peat warming and reflected only the response of heterotrophic processes to temperature, as photosynthetic and aerobic processes were excluded by the incubation design. The incubation results differ from the field, where dark CO$_2$ flux did not correlate with temperature treatment, possibly because autotrophic processes were excluded in the incubations, or because CO$_2$ production in the field was greatest at depths shallower than 20 cm, which were not included in the incubation experiments. The decreasing CO$_2$:CH$_4$ ratio with temperature in the surface incubations (Fig. 5) indicates that anaerobic respiration may become increasingly methanogenic with warming in agreement with results from previous incubation studies[18,19].

To further verify the role of surficial processes in the field CH$_4$ flux response, we compared the natural abundance $\Delta^{14}$C of the CO$_2$ (DIC) and CH$_4$ dissolved in peat porewater with the DOC and solid peat. DOC at S1 bog is younger than the peat at all depths[17], indicating that it is largely derived from recent photosynthate as opposed to the progressively older solid phase peat at depth (Fig. 6a). Increasing temperatures are likely to stimulate photosynthesis rates and increase root exudation of organic C available for decomposition[20]. The young age of the DOC and the lack of a temperature effect on DOC concentrations (Supplementary Fig. 4) show that there was not significant leaching of ancient catotelm C into the dissolved pool after 13 months of warming. The similarity between CO$_2$/CH$_4$ and DOC radiocarbon values is consistent with respiration fueled by younger surface-derived C sources, rather than by degradation of ancient catotelm C[14,15]. The difference in stable C isotope values of DIC and CH$_4$, represented by $\alpha_C$ [($\delta^{13}CO_2 + 1,000$)/ ($\delta^{13}CH_4 + 1,000$)], identifies shifts in the dominant methanogenic pathway, because hydrogenotrophic methanogens fractionate C more than acetoclastic methanogens[21]. The magnitude of the isotopic shift and the depth at which the shift occurred was similar across temperature treatments (Fig. 6b), suggesting that DPH did not significantly influence the depth

distribution of dominant CH$_4$ production pathways. This finding is contrary to what Dorrepaal *et al.*[22] and McCalley *et al.*[23] found following warming-induced permafrost thaw in peatlands and suggests that the response of heterotrophic respiration to climatic warming may differ between peatlands with different cryogenic histories, mineral contents, microbial population dynamics and plant community compositions[24]. In permafrost

settings—particularly syngenetic permafrost—the organic matter is frozen at a partially decomposed state, that is, decomposition is suspended preserving labile material. As permafrost thaws, that labile material becomes available, enhancing decomposition rates. In contrast, non-permafrost peatlands only experience seasonal freezing in surface peat, leading to millennia of slow decomposition of deep peat. In the case of S1 bog, over at least the time frame of this study, the decomposition of recalcitrant deep peat was not responsive to increasing temperature.

No correlation between microbial abundances and temperature treatment were observed (Fig. 7). However, both C decomposition and microbial community structure exhibited strong vertical stratification, similar to pre-treatment findings[16]. Members of the *Alphaproteobacteria* and *Acidobacteria* classes—which in peat contain abundant aerobic heterotrophs[25]—decreased in relative abundance with depth, whereas putative anaerobes (for example, *Deltaproteobacteria* and TM1) increased in the catotelm (Supplementary Figs 7 and 8). Microbial degradation of recalcitrant, lignin-like compounds that are abundant in peatland soils is mediated by the activity of extracellular oxidative enzymes, namely phenol oxidases and peroxidases[26,27]. These enzymes similarly showed no response to the DPH treatment (Supplementary Figs 11 and 12), although a clear vertical stratification in phenol oxidase activity occurred with peat depth, probably reflecting differences in dissolved $O_2$ availability.

No temperature effect was observed on the relative abundance of Archaea or methanogens (Supplementary Figs 9 and 10), further supporting the hypothesis that factors other than temperature are limiting decomposition in the deep peat. The majority of Archaea in the shallow peat were methanogens whose relative abundance declined with depth, in concert with methanogenesis rates (Supplementary Fig. 10). Correspondingly, the functional methanogen gene *mcrA* was more than ten times higher at depths 20–40 cm than at depths below 50 cm (Supplementary Fig. 9d), suggesting enhanced methanogenesis in the surface peat.

Although there is evidence of kinetic control on surface peat decomposition in our experiment, non-kinetic factors—such as chemical recalcitrance[17]—appear to be controlling the decomposition of deep C at S1 bog. Tfaily *et al.*[17] report a marked decrease in the o-alkyl C content of catotelm peat relative to acrotelm peat at S1 bog, indicating intensive decomposition of carbohydrates. Previous studies have linked o-alkyl C content to peat reactivity[28,29] and have observed clear decreases in o-alkyl peat content from northern peatlands to tropical peatlands (S. Hodgkins, personal communication). Thus, we hypothesize that the lack of reactivity of SPRUCE deep peat was due to the low o-alkyl C content of the soil organic matter. Therefore, future warming will probably have little effect on the conversion of catotelm C to $CO_2$ and $CH_4$. However, catotelm peat recalcitrance is a relative term. We have shown that catotelm peat is

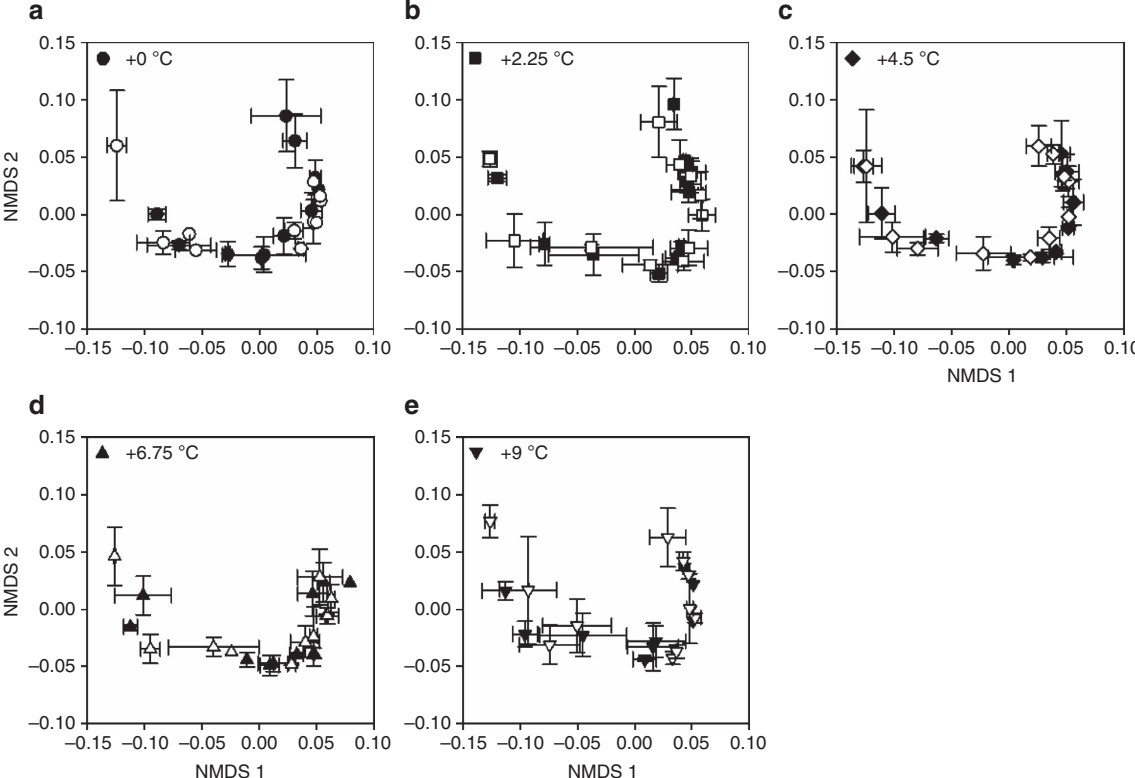

**Figure 7 | Microbial community structure in treatment plots.** Characterization of in situ microbial community structure by non-metric multidimensional scaling (NMDS) in (**a**) control plots, (**b**) + 2.25 °C plots, (**c**) + 4.5 °C plots, (**d**) + 6.75 °C plots and (**e**) + 9 °C plots. Symbols represent averages of two samples collected from each plot/depth within each temperature treatment plot from pre-DPH (2014, closed symbols) and 13 months post-initiation of the DPH (2015, open symbols) experiment. Error bars represent 1 s.d. of replicate results. Circles represent results from the control (0°C) plots, squares represent results from the + 2.25°C plots, diamonds represent results from the + 4.5°C plots, triangles represent results from the + 6.75°C plots, and inverted triangles represent results from the + 9°C plots. Final sequence data were normalized by cumulative sum scaling (CSS) and beta diversity indices were estimated based on Bray–Curtis and weighted as well as unweighted Unifrac distances. Significant differences in beta diversity were analysed by a PERMANOVA test on weighted Unifrac distance metrics with 1,000 permutations followed by Bonferroni correction of *P*-values.

recalcitrant with respect to temperature under its present conditions—water saturated, with fermentation and methanogenesis as the dominant organic matter decomposition processes. However, changes in any of those conditions—such as introduction of a favourable electron acceptor (for example, $O_2$)—could stimulate decomposition of the catotelm peat.

Other climate-induced perturbations to the ecosystem—changes in water-table depth, increased plant productivity, below-ground exudation of labile plant compounds or changes in plant communities—could have cascading effects on peatland C dynamics. Enclosure water table positions did not fluctuate due to DPH, indicating that the ecosystem responses we observed were driven solely by warmer temperatures. However, direct warming of surface peat is expected to lead to a lower water table depth, increasing $O_2$ availability throughout this soil horizon. Greater $O_2$ availability will enhance decomposition and aerobic $CH_4$ oxidation, probably resulting in an overall reduction of $CH_4$ emissions[30]. For example, lowering of the water table due to increased evapotranspiration could increase $O_2$ availability providing the necessary conditions for degradation of recalcitrant phenolic compounds in the catotelm, which have been proposed to protect the global C bank in deep peat through inhibition of microbial heterotrophy according to the 'enzyme latch' hypothesis[31]. However, recent studies have shown that temperature, water-table depth and perhaps even nutrient availability may control the strength of the enzyme latch, and that the response of phenolic compound degradation to climate drivers may be more complicated than originally hypothesized[32,33]. The lack of a legacy warming effect in the surface peat after the first 13 months suggests that the warming treatment did not have a lasting effect—relevant to $CO_2$ and $CH_4$ production rates—on the microbial community or the peat itself. Although peat decomposition was enhanced in incubations of surface peat, our results provide evidence that C decomposition in deep anaerobic peat is not kinetically constrained; therefore, peat decomposition is most likely to be thermodynamically limited by the absence of suitable electron acceptors.

It should be noted that the lack of response reported here may be specific to ombrotrophic bogs and does not necessarily reflect the expected or observed response from other peatland habitats such as fens or permafrost peatlands. However, even if global warming-induced increases in $CH_4$ production are confined to surface processes in ombrotrophic bogs, this could still represent a substantial natural feedback to anthropogenic climate forcing. Specifically, the exponential increase in $CH_4$ flux observed in the field plots coupled with the decrease in $CO_2:CH_4$ ratios in the surface peat incubations is troubling given that $CH_4$ has a sustained global warming potential 45 times that of $CO_2$ on a 100-year timescale[34]. Further, the surface responses reported here were likely to be underestimated due to energy losses that muted the warming treatment in surface peat. With the addition of surface warming, it is likely that the surficial response to temperature will be even greater. Thus, even if warming stimulates plant biomass production and enhances soil C sequestration, these effects are unlikely to completely offset the increases in $CH_4$ flux on this time scale. However, we must temper our interpretation, because the observed surface response may be a transient perturbation effect as has been seen in other climate manipulation experiments[35–37]. In addition, increased frequency and duration of low water table elevations and flow along near-surface lateral flow paths are most likely to affect surface peat, which may exacerbate or mitigate the responses that we observed[38,39]. For example, even with a temperature increase, a lowered water table could reduce $CH_4$ production, enhance oxidation and result in lowered $CH_4$ emissions. In peatlands, feedbacks exist among plant communities, water

table dynamics and physical properties of the peat resulting in a tight coupling between C and water cycling[35,40] that allows the system to self-regulate, resisting gradual environmental change until a catastrophic tipping point is reached and the system shifts towards a new steady state[13,35]. For example, *Sphagnum* and vascular plants alter environmental conditions such as light and nutrient availability, water table depth, temperature and pH[13,35]. The long-term SPRUCE experiment will enable us to examine whole-ecosystem warming, enhanced atmospheric $CO_2$ and water table feedbacks to these treatments, allowing us to clarify the internal mechanisms that control C cycling in a bog over a decade-long manipulative climate change study.

## Methods

**Site description.** The SPRUCE experimental site, S1 bog (8.1 ha), is located in northern Minnesota, USA within the Marcell Experimental Forest (N 47°30.476′; W 93°27.162′). The S1 bog has been the subject of extensive past research and has been described previously[10,16,17]. This precipitation-fed, ombrotrophic bog has an average pH of 4.1 at the surface that increases with depth to an average value of 5.1 at 2 m. Overstory vegetation is dominated by two tree species, *Picea mariana* (black spruce) and *Larix laricina* (larch), whereas the understory is composed mainly of low ericaceous shrubs, such as *Rhododendron groenlandicum* (Labrador tea) and *Chamaedaphne calyculata* (leatherleaf), as well as the herbaceous perennials *Maianthemum triflorum* (three-leaved Solomon's seal) and *Eriophorum vaginatum* (cottongrass). The bog surface is characterized by hummock and hollow microtopography, with *Sphagnum magellanicum* colonizing the hummocks and *Sphagnum angustifolium* the hollows. Typically, the hummocks are 10–30 cm higher than the hollows. Plant cover varies only slightly among the plots measured for surface $CO_2$ and $CH_4$ efflux. All plots have a nearly uniform cover of *Sphagnum* over the hummock–hollow complex over which an ericaceous shrub layer is present.

**Deep peat heating.** The SPRUCE project involves an ecosystem-scale climate manipulation in the S1 bog. The experimental design includes ten 12 m-diameter enclosures that are warmed to five temperatures ( $+0$, $+2.25$, $+4.5$, $+6.75$ and $+9\,°C$), with duplicate plots to be subjected to ambient and $\sim +500$ p.p.m.v. $CO_2$. In the most novel aspect of this experiment, the peat is warmed throughout the peat column to depths of 2–3 m (ref. 6), providing the first field-scale examination of the responses of deep peat to climate forcing. Briefly, low-wattage, 3 m-long below-ground heaters were installed equidistant around the circumference and beneath each treatment plot, to heat the soil to the desired temperature differential[10]. The open-top enclosure design allows surface warming and enhancement of atmospheric $CO_2$, whereas sub-surface corrals hydrologically isolate each experimental plot and allow for changes in water table associated with warming and elevated $CO_2$ to develop. However, DPH was the only experimental treatment applied during this study.

DPH was initiated between 17 June and 2 July 2014 as the electrical systems for each plot became available. Stable target treatment temperature differentials at 2 m deep were achieved in all plots by early September 2014. DPH is accomplished by an array of 3 m vertically installed low wattage (100 W) heating elements housed within plastic-coated iron pipes and placed throughout the plots in circles of 48, 12 and 6 heaters at 5.4, 4 and 2 m radii, respectively. A single heater was also installed at the plot centre. Exterior heaters in the circle of 48 apply 100 W across the full linear length of the heater and all interior heaters apply 100 W to the bottom one-third of each resistance heater (pipe thread core heaters, Indeeco, St Louis, MO). DPH within the experimental plots is achieved through proportional-integral-derivative control of three exterior (the circle of 48 split into alternating thirds) and two interior circuits of the resistance heaters. The reference depth for temperature control is 2 m deep.

Temperature differentials within a treatment pair were typically within 0.5 °C of the target temperatures throughout the measurement period. Temperature variation in the no-energy-added control plots was likely to be driven by differences in tree canopy cover with the greater cover leading to warmer peat temperatures (that is, less heat loss to the sky). Once deep peat temperature differentials were achieved, they were largely maintained from 1 to 2 m deep during large seasonal shifts in temperatures (Fig. 2).

**Analysis of $CH_4$ and $CO_2$ flux.** Measurement of $CO_2$ and $CH_4$ emissions from the peatland was conducted in 1.2 m diameter permanent collars embedded 10 cm into the peat[41]. Briefly, collars were covered with an opaque dome under which headspace accumulation techniques were applied. Gas accumulation under the darkened dome was measured with open path $CO_2/H_2O$ (LiCor 7500) and $CH_4$ analysers (LiCor 7700). An individual observation lasted only minutes without dramatic changes in temperature, pressure or target gas concentration above the surface of the peat. Seasonal flux measurements were fit against the average

temperature from 1 to 2 m below the hollow surface with an exponential regression model using SigmaPlot v 12.3 and significant relationships identified at $P < 0.05$.

**Analysis of $CH_4$ and $CO_2$ production in anaerobic incubations.** Intact soil cores were collected at 20–30, 50–75, 100–125, 125–150 and 175–200 cm depths from each experimental plot in September 2014 and June 2015, after $\sim 4$ and 13 months of DPH, respectively, to discern how rates of $CO_2$ and $CH_4$ production varied with depth. All depths were measured relative to the surface of the hollows. To prevent compression of surface peat samples, a serrated knife was used to collect a 10 cm-diameter core from the hollow surface to $\sim 20$ cm within the peat profile. A 5 cm-diameter Russian corer was subsequently used to extract the remaining samples up to 2 m deep. The soil cores were kept anaerobic, stored on ice and shipped overnight to the University of Oregon, where incubations commenced immediately within $1 °C$ of *in situ* temperatures. Samples were slurried with a 1:1 mixture of peat and porewater collected from the same plot and depth. $CO_2$ and $CH_4$ production concentrations were determined[42] during the course of the 10-day incubation.

All statistical analyses were conducted using SPSS Statistics version 22. Data were tested for normality and log-transformed where the transformation resulted in a significant improvement in overall distribution. General Linear Model analysis was used to investigate the effect of temperature, depth and the interaction of these two variables on $CH_4$ and $CO_2$ production, as well as the $CO_2$:$CH_4$ ratio. If significant differences among depths were detected ($P < 0.05$), pairwise comparisons using Tukey's honest significant difference test ($P < 0.05$) were conducted. If not significantly different, depths were combined for linear regression analysis. If normally distributed, $CH_4$ and $CO_2$ production rates and $CO_2$:$CH_4$ ratios were combined across sampling time points and linear or exponential regression was used to determine the temperature response of each process.

**Analyses of porewater gas and isotopic composition.** Porewater samples were collected in June 2015 for analysis of $CH_4$ and $CO_2$ concentrations, $\delta^{13}C$ and $^{14}C$ using permanently installed piezometers at 25, 50, 75, 100, 150 and 200 cm depths within each experimental plot. Piezometers were covered, but not sealed, when not being actively sampled. The diameter of the piezometers was $< 1$ cm, which limited oxygen diffusion, and piezometers tubes were pumped dry 24 h before sampling, to ensure that the sampled water was not in prolonged contact with the atmosphere before sampling. Surface water samples were collected using perforated stainless steel tubes that were inserted in the peat to 10 cm or the top of the water table, whichever was shallowest. Pore water was immediately filtered to 0.7 μm in the field using Whatman glass-fibre filters, then stored in pre-evacuated glass vials sealed with butyl stoppers. Phosphoric acid (1 mL of 20%) was added to each sample, to preserve for shipment to Florida State University. Samples were analysed for $CH_4$ and $CO_2$ concentrations and stable isotopic composition ($\delta^{13}C$) on a ThermoFinnigan Delta-V Isotope Ratio Mass Spectrometer using the headspace equilibration method with He. Each sample was analysed twice and the average results for each sample were recorded. Analytical precision was 0.2‰.

Preparation of $\Delta^{14}C$-DOC, $\Delta^{14}C$-DIC and $\Delta^{14}C$-CH$_4$, and $\Delta^{14}C$ peat samples was done at Florida State University. DOC was freeze dried in combusted 9 mm Pyrex glass tubes. Oxidizing agents, cupric oxide, copper shots and silver, were added and the tubes evacuated and flame sealed on a vacuum line. The sealed tubes were then combusted at 580 °C for 18 h to convert the organic carbon to $CO_2$ gas[43]. Following combustion, the produced $CO_2$ was taken back to the vacuum line, cryogenically purified and sealed into 6 mm glass tubing. $\Delta^{14}C$-DIC and $\Delta^{14}C$-CH$_4$ were prepared by He stripping and subsequent combustion (for CH$_4$ and cryogenic trapping). The 6 mm tubes for $\Delta^{14}C$ analysis were sent to National Ocean Sciences Accelerator Mass Spectrometry Facility and the Lawrence Livermore National Laboratory for analysis.

Pore water samples for measurement of total organic carbon (TOC) concentrations were collected every 2 weeks beginning in late August 2013 and continuing throughout the DPH experiment. These samples were collected from a set of 5 cm internal diameter PVC piezometers installed in each experimental plot. The piezometers had 10 cm screened intervals that opened at depths of 0, 30, 50, 100, 200 and 300 cm. Water was pumped using a peristaltic pump via flexible sections of Salastic and silicon tubing that was attached to a static 0.6 cm internal diameter PVC tube inside each piezometer (a design that was intended to reduce contamination via intermittent tube insertion into piezometers for sampling). Samples were collected in 250 ml low-density polyethylene bottles that were chilled and transported to the Forestry Sciences Laboratory of the USDA Forest Service. Samples were then refrigerated until analysed, typically 1–4 days after collection for TOC concentration. TOC concentration was measured on unfiltered water samples using the non-purgeable organic carbon method on a Shimadzu TOC-VCP using Standard Method 5310 B[44] (equivalent to EPA 215.1). The method detection limit was $0.5\ mg\ l^{-1}$ for TOC concentration.

**Microbial community analyses.** Intact soil core samples were collected from 11 depth intervals (0–10, 10–20, 20–30, 30–40, 40–50, 50–75, 75–100, 100–125, 125–150, 150–175 and 175–200 cm) at each of the 10 SPRUCE experimental plots in June 2014 and June 2015, before and 13 months into DPH, respectively, to elucidate the microbial community response to warming. Soil samples were frozen

immediately and shipped on dry ice to the Georgia Institute of Technology, where they were stored at $-80 °C$ until analysis. Total DNA was extracted from homogenized peat samples with the MoBio PowerSoil DNA extraction kit (MoBio, Carlsbad, CA) according to the manufacturer's protocol followed by cleaning with the MoBio PowerClean Pro DNA Cleanup Kit (MoBio). Abundance of bacterial, archaeal and fungal populations were determined by quantitative PCR using primers targeted to amplify their respective SSU rRNA genes[45–48] and the *mcrA* gene was targeted to assess the methanogen population[49]. Reactions were performed in triplicate on a CFX96TM Real-Time PCR Detection System (Bio-Rad Laboratories) with iQ SYBR Green Supermix (Bio-Rad, CA, USA) using previously described standards and conditions (Supplementary Table 1). DNA extractions were quantified with the Qubit HS assay (Invitrogen) and 20 ng per reaction was applied. The diversity and composition of prokaryotic communities was determined by applying a high-throughput sequencing-based protocol that targets PCR-generated amplicons from V4 variable regions of the 16S rRNA gene using the bacterial primer set 515 F (5′-GTGCCAGCMGCCGCGGTAA-3′) and 806R (5′-GGACTACHVGGGTWTCTAAT-3′)[50]. Amplicons were barcoded with unique 10-base barcodes (Fluidigm Corporation) and sequencing was conducted on an Illumina MiSeq2000 platform at the Research Resources Center at the University of Illinois at Chicago following standard protocols[51,52] (http://www.earthmicrobiome.org/emp-standard-protocols/16s/). The generated sequence data are available from the National Center for Biotechnology Information at SRP071256.

**Sequence processing and analysis.** Initially Illumina-generated 16S rRNA gene sequences were paired with PEAR[53] and primers were trimmed with the software Mothur v1.36.1 (ref. 54). Resulting sequences were quality filtered using a Q30 minimum and processed using the standard QIIME 1.9.1 pipeline[46,47]. Sequences were clustered into operational taxonomic units with a threshold of 97% identity. Chimeric sequences, identified by ChimeraSlayer, chloroplast, mitochondria, singletons, unclassified and eukaryotic sequences were removed from the final data. Taxonomies of these high-quality sequences were assigned via the greengenes database using the RDP classifier[55] with a minimum confidence threshold of 50%. Sequences of known methanogens were extracted from all sequences according to recent methanogen databases[56,57]. The sequences that did not match any taxonomic Class were also removed. Taxonomic-based alpha diversity was calculated using the total number of phylotypes (richness) and Shannon's diversity index ($H'$). Faith's phylogenetic diversity was calculated to assess phylogenetic-based alpha diversity. Final sequence data were normalized by cumulative sum scaling[58] and beta diversity indices were estimated using Bray–Curtis and weighted as well as unweighted UniFrac distances[59,60]. Significant differences in beta diversity were analysed by a permutational multivariate analysis of variance test with 1,000 permutations followed by Bonferroni correction of $P$-values. To determine changes in microbial community composition, results from core sections were grouped based on beta-diversity groups (0–10, 10–30, 30–75 and 75–200 cm) and significant differences between years (pre and during heating) were assessed with a Mann–Whitney test.

**Enzyme activities.** Enzymatic assays were performed following published microplate protocols[46,61]. Peat suspensions from seven core sections (0–10, 10–20, 20–30, 30–40, 40–50, 50–75 and 75–100 cm) were prepared by homogenizing 2 g of peat in 20 ml of 50 mM acetate buffer (pH 4.0). Peat homogenates were centrifuged for 5 min at 5,000 $g$ and clear supernatants were used for measurements of phenol oxidase and peroxidase activities. Enzymatic activities were measured by combining 1 ml of clear peat suspension with 1 ml substrate solution (10 mM 2,2′-azino-bis (3-ethylbenzothiazoline-6-sulphonic acid) in 50 mM acetate buffer, pH 4.0). For measurements of peroxidase activity, peat suspensions were diluted 1:20 with 50 mM acetate buffer (pH 4.0) and reaction was initiated by adding 80 μl of 0.3% hydrogen peroxide. Assays were incubated for 12–24 h at room temperature. Enzymatic reaction propagation was monitored spectrophotometrically at 420 nm. Maximal reaction rates were calculated from linear reaction stage and expressed as μmol or mmol $h^{-1} g^{-1}$ wet peat for phenol oxidase or peroxidase, respectively. Statistically significant differences between years were determined by Student $T$-test.

**Data availability.** All data presented in this manuscript and the Supplementary Information files, including figure source data, are publicly available from the SPRUCE long-term repository (doi: http://dx.doi.org/10.3334/CDIAC/spruce.026).[62]

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

## Acknowledgements

This material is based upon work supported by the U.S. Department of Energy, Office of Science, Office of Biological and Environmental Research. Oak Ridge National Laboratory is managed by UT-Battelle, LLC, for the U.S. Department of Energy. Funding was provided by the U.S. Department of Energy under contract number DE-AC05–00OR22725. Work conducted by J.E.K., M.K., J.P.C. and R.M.W. was supported by contract number DE-SC0012088, and by A.M.H., L.P.-M., C.M., J.K.K. and S.D.B. by contract DE-SC0008092 from the Office of Biological and Environmental Research, Terrestrial Ecosystem Science (TES) Program, U.S. Department of Energy.

## Author contributions

R.M.W. and A.M.H. wrote the manuscript with contributions from all coauthors. R.M.W. and J.P.C. collected and analysed pore water data. R.M.W., J.P.C., K.J.M. and T.P.G. collected and analysed radiocarbon samples. A.M.H., J.K.K., S.D.B. and C.M. designed the incubation experiments. A.M.H. conducted incubations with C.M. and analysed resultant data with S.D.B. C.M., R.K.K., L.P.-M., C.W.S., L.A.K., J.E.K., J.K.K. and M.M.T. assisted in coring events. S.D.S. and N.A.G. collected and analysed DOC samples. P.J.H. collected and analysed flux measurements, and manages all field activities with R.K.K. J.E.K., M.K., C.W.S. and L.A.K. designed, collected and analysed microbial community data. M.K.K. and J.E.K. designed the enzyme activity experiments, with M.K.K. conducting the analyses.

## Additional information

**Competing financial interests**: The authors declare no competing financial interests.

**How to cite this article**: Wilson, R. M. *et al.* Stability of peatland carbon to rising temperatures. *Nat. Commun.* **7**, 13723 doi: 10.1038/ncomms13723 (2016).

**Publisher's note**: 

