## [Peer Review File · Nature Communications]

Reviewers' comments:

Reviewer #1 (Remarks to the Author):

This study investigates the effect of warming 'at depth' on peatland carbon processes, decomposition and surface fluxes. Results are generated by a unique experimental set up, where large soil monoliths in situ were heated at depth.

This scientific question under investigation, namely the vulnerability of the northern peatland C pool to climatic warming, has been a subject of interest for some time and as the authors clearly point out is significant for climate change because of the immense size of this C pool and is still highly uncertain.

The main finding of this research is that warming of the peat soil at depth does not increase decomposition in these deeper peats nor does it lead to an increase in surface CO₂ flux and although surface CH₄ flux at the surface increases it is due to surface processes and not heating of the deep peat. This result is quite novel and important. The fact that these findings contradict what has been found in permafrost peatlands (where deep C stores are sensitive to warming) is also novel and could prompt new avenues of research. Although the findings are well supported by multiple lines of research, making the case compelling, there are a few issues that should be considered.

1. There is no information about water table depth (WTD) in the paper or supplemental materials, apart from a single statement that WTD was allowed to vary with the heating manipulations. Since the processes of decomposition, production and transport of the C gases are intimately tied to WTD it would be important to know what happened to WTD in relation to the results presented.
2. Somewhat related to the above point - the term "peat surface" referred to in the manuscript is somewhat vague, presumably it means a layer between the top of the moss and some depth, is it the acrotelm or something else. A short discussion or clarification would help.
3. In relation to Extended data Figure 1, the statement "In the absence of air warming during this phase of the experiment, anticipated energy loss at the surface resulted in less warming of the surface peat", which occurs in both figure caption and the manuscript body (pg. 2 lines 21-24) seems odd as the data mostly show warmer temperatures at the surface of the peat in this diagram. This is likely true in longer term average temperatures, but not in this figure.
4. Some comment on the realism of this experimental design is needed. It is understandable why the experiment was set up the way it was, but i) warming of peat by climate change will not happen from below, heat must be transferred down the peat profile to the deeper peats, and ii) the manipulations >+2 degrees C are likely far beyond what we can expect to occur at this depth in the peat with even the most liberal of climate change estimates for the next century. So, what does this experiment say about the real future?
5. Minor point, but final line of the manuscript (pg. 6 lines 7-9) is a bit self-serving and likely not needed.

Reviewer #2 (Remarks to the Author):

This study presents results on the fate of peat carbon stocks from a novel experiment in which peat was warmed in situ to a depth of 2 m. The authors investigate the decomposition of the organic matter reservoir using a variety of methods including surface fluxes of CO₂ and CH₄, laboratory incubation of CO₂/CH₄ production, microbial community analysis, dissolved carbon pools and ages, etc. Overall, the various lines of evidence suggest that warming of the peat is unlikely to result in organic matter oxidation, but may shift the balance towards higher CH₄ flux. Given the large peat stock of C and potential for a feedback to climate change, the findings are of broad interest and deep warming of peat is novel.

There are a few inconsistencies in the data interpretation that need to be addressed prior to publication (e.g. statement of no clear temperature response for deeper peat in incubations,

although there is clearly an increase from the figures and then this response varies compared to the field fluxes), but overall the methods and interpretations are sound. The one thing I would suggest to add is a short discussion on mechanisms for the protection of peat from increasing rates of decomposition under warming temperatures (the why?) even if the answer is still unclear. This will help to push the research forward by identifying what is still unknown.

Finally, the authors have observed increases in CH₄ flux and a shift to lower CO₂:CH₄ ratio in the lab incubations. They make some statements at the end of the paper about the importance of this to GWP of peatland in response to deep warming, suggesting that even if the C stock is stable, peatlands may become a GHG source. I suggest that author should carefully consider how confident they are with this finding is and whether or not it should be included in the abstract (currently it is not). They end the paper with it, highlighting its importance, so I would suggest that it should be included in the abstract in this case.

Specific comments:

Page 1, line 29: "rapid" can mean different things to different people. Hundreds of years can be considered rapid on a geological time scale. Can you be more precise here? Also, don't you think the results would warrant a strong statement here - possibly "changes to the large reservoir are very unlikely" in place of "may be minimal"?

Page 2: Do you need the heading "Introduction". There are no other headings and the text here is actually introduction, results and discussion.

Page 2, line 1: As the deeper peat is relatively well insulated by the surface peat, how much warming is really expected at depth, particularly down to 2 m? I agree it will eventually warm in response to the warming atmosphere but won't this happen very slowly? Is the warming induced in the experiment representative of climate change?

Page 2, line 4: Is reference to Hanson et al. 2011 really the best here. That study tests methods to heat deep soil, but does not determine that deep heating will occur in response surface heating

Page 2, lines 22-23: So, without the surface heating the temperature profile is not really realistic due to the heat loss at the surface. Does this have implications for the interpretation of the results? Since incubations suggest the 20 cm depth was most impacted by the increase in temperature, the observed temperature profile may have limited the impact on surface fluxes compared to a profile warmed uniformly. Please comment.

Page 2, lines 28-29: What happened to the plant community in response to heating? Plant respiration can dominate ecosystem respiration during the summertime, and changes in plant cover might mask differences in soil respiration.

Page 3, lines 21-22: Do you think this is a fair statement? The relationship between CO₂ production and temperature was not significant, but there appears to be a clear increase after 13 months

So what is protecting the peat from decomposition? Recalcitrance? Anoxia? Build-up of end-products? The incubations suggest you can release of this as CO₂ production does increase, but it doesn't happen in the field...

Page 3, lines 23-24: Has anyone else observed this change in ratio with temperature? There have been a lot of peat incubations, so I'd be surprised that there is nothing you can cite here.

Page 5, Lines 24-27: This is a pretty big however. If the deeper peat is exposed to different conditions due to water table drawdown or the oxidation of the surface layers and exposure, will it still respond in this way, or will the "protection" be gone. This is pretty critical for concluding that

changes in the deep peat C will be minimal. I agree with you, but maybe it is not as clear cut.

Page 9, lines 13-14: What do you mean here. The seasonal flux was fit to an exponential regression relating it to what?

Page 9, line 17: How were the intact cores collected and how large were they?

Page 9, line 24: How long did the incubations last?

Page 10: lines 7-8: Were the piezometers sealed? Was there any exposure of the porewater at depth to oxygen? Clarify in text.

Page 11, line 3: There's an extra space here in the word "sample"

Extended data figure 2: For CO₂ flux, it almost looks like 2 relationships here with a break at around 9 degrees. Would you expect CO₂ flux to be linked to deep (1-2 m) soil temperature given that a lot of the source is the near surface oxic peat (and the plants in the summer)? Is there a relationship with surface peat temperature? And again, I think the CO₂ flux is less easy to interpret because it includes the plant respiration, which is likely variable between plots and might mask some of the patterns. Can you comment on how comparable the plant cover was between the various plots and different treatment temperatures?

Extended data Figure 5: There is definitely higher CO₂ production after warming at all depths, but then no legacy effect. What do you think the cause is? Is this a more active microbial community or a change in substrate quality or both? Why is it not reflected in the surface flux?

Reviewer #3 (Remarks to the Author):

As the apparent initial phase of a long-term ecosystem manipulation project, Wilson et al., investigated how CO₂ and CH₄ fluxes varied following 1 year of subterranean (~ 2 m) heating (0.0oC to 9oC above ambient) of a peatland. Through surface level measurements of CO₂ and CH₄ evolution at plots subjected to varying degrees of warming the authors observed that CH₄ fluxes from the surface increased. However, based on in situ measurements of community composition, enzymatic activity, and isotopic enrichment analyses along with laboratory incubations of peat from the surface and the deeper biosphere they determined that this flux was due to turnover of peat at the surface and not the deeper (> 1 m) carbon pool.

Overall, this is a well written and presented study addressing a very important questions. Specifically, since peatlands contain a substantial portion of terrestrial carbon therefore it is important to predict how the fate of this carbon might be affected due to climate change. However, there are two major criticisms that cause this manuscript to fall short of demonstrating what this statement at the end of the abstract "Our results indicate that rapid changes to the large reservoir of deep buried C in peatlands may be minimal under future climatic warming."

Major concerns:

1-- the experimental design used here seems to limit the relevance of this data to the larger ecosystem. In particular, the authors do not establish the validity of heating below ground without corresponding manipulations to the surface. This is important since I am unaware of any global warming models where heating originates from the deep subsurface and radiates upward. By conducting the ecosystem manipulation in this manner, they effectively limit the impact of their study as it poorly accounts for potential, if not likely, changes to the ecosystem that may originate at the surface which could have cascading effects that reach the deeper parts of the soil column. The authors mention this to a degree (pg. 5 ln 27 - 31), but it doesn't seem to be taken into

account in the experimental design. Thus, the results from this study seem to be more of a baseline observation on the limiting effects temperature alone has on the deep carbon pool, if nothing else is changed. This is certainly of some interest given the importance of the problem, however, the relevance to native ecosystem processes would need to be established to have broad appeal to the scientific community.

2--The authors conducted several experiments to both localize methane production in the column and assess possible reasons why deep carbon wasn't affected, with consistent results: temperature did not appear to affect microbial deep composition, enzymatic activity or greenhouse gas production from the deeper soil column. However, a second major criticism is that in spite of the multiple incubations and measurements conducted, the study lacks a few experiments that are needed to help solidified the cause of this persistence and thus strengthened this study. Specifically, is the deep peat inherently more resistant to microbial decomposition or is this persistence due to the microbial composition or the environmental conditions deeper in the profile (e.g. low pH, oxygen limitation)? If it is actually abiotic/biotic conditions deeper in the profile that are the controls, not the peat itself, prolonged surface warming may alter the abiotic/biotic factors. While they do mention this to some degree on page 5 line 23, additional incubation experiments such as incubation of this older peat with populations or conditions more similar to the surface environment would be required to establish that it is the properties of the peat that limit decomposition. This likely will reveal, as the authors suggest as a possibility, that oxygen concentration is limiting deep peat mineralization, which would be an important finding. However, if this deep peat is truly resistant to microbial decomposition as a result of its chemical structure and not other biotic or abiotic factors, this would also be of high interest.

Minor points:

- The exponential regression in figure 1a looks to be a poor fit to the data ($r^2 < 0.14$ in both cases) and therefore it is unclear why this is shown or described as an exponential relationship (pg 2., lines 25 - 26).
- Please be consistent with labeling figure and extended figure captions in the descriptions. In some cases the label (i.e., 'a','b','c') is at the beginning of the description and at other times it is at the end.
- Pg 4, line 22 - what is the difference between the 'depth' and 'magnitude' of a shift?
- Pg 3, last paragraph - if there were not consistent effects observed due to temperature changes, why would you expect to see 'legacy effects'?
- A schematic drawing of the site would be helpful in the SI.
- Extended data, figure 8. Why was only the top 2 cm examined?

Reviewer #4 (Remarks to the Author):

I think the main finding of the non-responsiveness of deeper peat to even dramatically increased peat temperature is a very important finding. This finding, if it can be universally applied, is very good news for the planet: it means that a significant component of what made up the proposed northern "carbon bomb" is not a "bomb" but more like a fire cracker, and a pretty small one at that. This finding is well supported by the several lines of evidence the authors present. I am very impressed with how tight the authors' arguments are and the varying lines of evidence they bring forward to support the case. I would say they have pretty much nailed it on this part of the manuscript. They are also careful to differentiate their findings from those of others who have worked on deeper peats in permafrost peatlands. I do think it is important for the authors to add a sentence to emphasize their results apply to ombrotrophic bogs and do not necessarily apply to transitional or blanket bogs, or mineral poor to mineral rich fens. Their results may apply to some

cases but they provide no evidence or argument as to why they should apply. Why this is important is a significant amount of carbon stored in peatlands is stored in other systems than bogs. For example, if we look at the Canadian peatland statistics about 70% of peatlands are bogs and this represents about 130 GT C in Canada alone, but the remaining peatlands are mostly fens (~30 Gt C in Canada). I am less impressed with the significance of the finding of increased methane emissions from the near surface peat, or the lack of response in HR. The authors results are interesting, but I am concerned about the treatment design too put much as much stock in these results. Firstly, it is a manipulation experiment and the authors have subjected their peatlands plots to a very large and relatively sudden temperature increase. From the results of other manipulation experiments we know there can be a very large initial change and then over time the change becomes less. The transient nature of the response of ecosystems early in the treatments make them very hard to interpret. Many of the authors on this manuscript have done other manipulation experiments and it is their papers that show the initial large response and then a subsequent lesser response. I think the authors need to acknowledge at minimum what happens to ecosystems when they are pushed rapidly far from equilibrium. The second concern is that of hydrology. I have discussed the design of SPRUCE with several of the authors so they will not be surprised with this concern. The results of the deeper peats is not influenced significantly by the hydrology but the surface peats are – i.e. there is no way the water balance and energy balance of a bogs (in my experience through both measurements and modelling) will change the moisture contents much below a metre depth. However, I would expect the moisture contents to change significantly in the surface peats with temperature changes in the order used in this study. However, the experiment does not explicitly experiment with changes in moisture content or lowering of a water table. If the authors look at the literature on fluxes of CO₂ and CH₄ from bogs they are quite responsive to variations in temperature, water table depths and/or soil moisture in the acrotelm. Later in the paper the authors acknowledge this limitation but this could come much earlier as I think it should temper and qualify the evidence presented on the response of near surface peat. For example, I would anticipate with such temperature increases there would have to be a commensurate decrease in acrotelm SMC and a corresponding increase in CH₄ oxidation. Are the authors results only providing part of the story (temperature response) while constraining another part of the story (moisture)? I am not arguing the authors' results are not interesting, they are but they confirm the results of many previous laboratory studies. Those same laboratory incubations (again some done by the authors of this paper) also show the important of changes in moisture. I suggest the authors need to show more caution in the interpretation and significance of the changes, or lack thereof in the shallow peat. I think it is critical that they be much more cautious on the universality of those results and much more self-critical on the design of their treatments for assessing the responses as representative of what one would expect in natural systems. I think this can be handled very easily – emphasize the deep peat results and de-emphasize the near surface peat results, and add in some qualifications. My comments below related to specific parts of the manuscript but largely reflect the general comments above. I definitely think the lack of response of deeper peat is worthy of publication in Nature. Pg 1 Ln 29 A critical finding and one that is global significant. The only concern is will this relative insignificant response continue. I am always concerned about short term responses from manipulation experiments. Pg 2 Ln 14 An important constraint that the SPRUCE team is well aware of is that deep warming is likely to occur with significant changes in moisture contents - i.e. peatlands are likely to become drier at the same time as they become warmer. While SPRUCE is not designed to test the effects of reduced WTDs and reduction in water contents in the

unsaturated zone, this constraint should be recognized in the interpretation of the authors' results. What would be the result if the mean WTDs were to be reduced say 10 to 30 cm and the SMC were to decrease in the unsaturated zone by several ten percent? The results for the surface peat is what I would expect based on numerous incubations but the in situ measurements are an important addition. However, the incubation experiments that have also included moisture change show the results from SPRUCE are different. The results for the deeper peats are not what I expected though are within the realm of possible results I would have anticipated. This is where the authors' results are unique and I think substantive in nature. First they show little response, which does vary with the other results reported in the literature, but more importantly the authors can convincingly explain what they got the results they did. That is, they provide the process level data to substantiate the system level response – or more correctly the lack of response of the deeper peat.

Pg 3 Ln 7 This makes the issue of moisture change even more important to consider since it will be in the surface layers where the response seems to occur. From continuous EC measurements of CH₄ from bogs like Marcell, there tends to be a relationship with moisture contents albeit a complicated one (e.g. see Brown et al. 2014 but there are many more). Brown, M. G., E. R. Humphreys, T. R. Moore, N. T. Roulet and P. M. Lafleur (2014). "Evidence for a nonmonotonic relationship between ecosystem-scale peatland methane emissions and water table depth." *Journal of Geophysical Research: Biogeosciences* 119(5): 2013JG002576.

Pg 4 Ln 15 Does the warming increase photosynthesis - i.e. the supply of exudates, and/or does the warming alter the production of more labile components of the exudates? What is the sources of the exudates? Pg 4 Ln 27-28 An important contrast! Could it be that the deeper peat in the permafrost setting is nowhere as recalcitrant as it is in Marcell? After all the peat is locked into the permafrost at ~ 50 to 70 cm. The work by Zimov and others on yedema, which I know is very different, has shown the C in permafrost when thawed supports a consider amount of heterotrophic respiration.

Pg 5 Ln 15-17 The reason I raise the moisture issue. Pg 5 Ln 21-23 This is the most important finding! The surface responses are interesting but may be highly transient. I think it would be a stretch to argue (I don't think the authors are arguing this) that we should expect continued elevated CH₄ from the surface layers or a lack of response of CO₂. This response may be very short term? The SPRUCE experiment should be able to answer this as long as the structural design of the SPRUCE does not inhibit the self-regulation between NPP and decomposition that exists in bogs. This is a longer regulation than the duration of SPRUCE. The authors should recognize that they are dealing with self-regulating ecosystems and this will have a profound effect on the difference between short-term and long-term responses. In addition, the design of SPRUCE may actually interfere with the ability of the ecosystem to self-regulate. I would have to think this through and possibly run some simulations with a model such as HPM (Frolking et al. 2010) or DigiBog (Morris et al. 2012, 2013, 2015) to see what the impact of restricting lateral water flow. Lateral water flow is a critical element to the self-regulation of peatlands (see Belyea & Baird 2006 or papers by Eppinga et al. 2009). I am not arguing the authors work on self-regulation but I do think they need to acknowledge that the ecosystem they are working on has this relative unique characteristics compared to most other ecosystems. In very wet and dry ecosystems (arid lands) water plays a far more critical role than it does in other ecosystems.

Belyea, L. R. and A. J. Baird (2006). "Beyond "the limits to peat bog growth": cross-scale feedback in peatland development." *Ecological Monographs* 76: 299-322. Eppinga, M. B., P. C. De Ruiter, M. J. Wassen and M. Rietkerk (2009). "Nutrients and hydrology indicate the driving mechanisms of peatland surface patterning." *American Naturalist* 173(6): 803-818. Eppinga, M. B., M. Rietkerk, M. J. Wassen and P. C. De Ruiter (2009). "Linking habitat modification to catastrophic shifts and

vegetation patterns in bogs." *Plant Ecology* 200(1): 53-68. Frohking, S., N. T. Roulet, E. Tuittila, J. L. Bubier, A. Quillet, J. Talbot and P. J. H. Richard (2010). "A new model of Holocene peatland net primary production, decomposition, water balance, and peat accumulation." *Earth Syst. Dynam. Discuss* 1: 115-167. Morris, P. J., A. J. Baird and L. R. Belyea (2012). "The DigiBog peatland development model 2: Ecohydrological simulations in 2D." *Ecohydrology* 5(3): 256-268. Morris, P. J., A. J. Baird and L. R. Belyea (2013). "The role of hydrological transience in peatland pattern formation." *Earth Surf. Dynam.* 1(1): 29-43. Morris, P. J., A. J. Baird, D. M. Young and G. T. Swindles (2015). "Untangling climate signals from autogenic changes in long-term peatland development." *Geophysical Research Letters* 42(24): 10788- 10797. Pg 5 Ln 27-31 I do not like promoting our own work in reviews but there are a series of papers by Andrew Pinsonneault et al. that have just come out that the authors might want to look at. Chris Evans was Drew's external examiner (Evans is a coworker of Freeman's - the author of the ELH) and he thinks there is more to it than simply the ELH. Pinsonneault, A. J., T. R. Moore and N. T. Roulet (2016). "Temperature the dominant control on the enzyme-latch across a range of temperate peatland types." *Soil Biology and Biochemistry* 97: 121-130. Pinsonneault, A. J., T. R. Moore, N. T. Roulet and J. F. Lapierre (2016). "Biodegradability of VegetationDerived Dissolved Organic Carbon in a Cool Temperate Ombrotrophic Bog." *Ecosystems*: 1-14. Pg 6 Ln 1-3 This is where I am concerned because it is where water is critical. For example the net ecosystem carbon balance of MB, the longest record for a bog (now 18 years) is correlated with mean growing season WTD as well as growing season temperature. We have presented this at conferences but the manuscript is still in preparation. Pg 6 Ln 5-7 This is again where the changes in moisture become critical. Nigel Roulet, McGill University, June 2016

Reviewer #1 (Remarks to the Author):

This study investigates the effect of warming 'at depth' on peatland carbon processes, decomposition and surface fluxes. Results are generated by a unique experimental set up, where large soil monoliths in situ were heated at depth.

This scientific question under investigation, namely the vulnerability of the northern peatland C pool to climatic warming, has been a subject of interest for some time and as the authors clearly point out is significant for climate change because of the immense size of this C pool and is still highly uncertain.

The main finding of this research is that warming of the peat soil at depth does not increase decomposition in these deeper peats nor does it lead to an increase in surface CO₂ flux and although surface CH₄ flux at the surface increases it is due to surface processes and not heating of the deep peat. This result is quite novel and important. The fact that these findings contradict what has been found in permafrost peatlands (where deep C stores are sensitive to warming) is also novel and could prompt new avenues of research. Although the findings are well supported by multiple lines of research, making the case compelling, there are a few issues that should be considered.

1. There is no information about water table depth (WTD) in the paper or supplemental materials, apart from a single statement that WTD was allowed to vary with the heating manipulations. Since the processes of decomposition, production and transport of the C gases are intimately tied to WTD it would be important to know what happened to WTD in relation to the results presented.

We monitored WTD in all 10 plots and several ambient plots during DPH (June 2014 and beyond). As we observed only modest heating of the surface, we did not expect, nor did we see, differences in WTD due to heating. As expected, water tables varied with rainfall inputs, snowmelt inputs, and

evapotranspiration – all processes that were not influenced by deep heating. The lack of a DPH effect on WTD during the DPH measurement period strengthens our temperature response arguments. That is, in the absence of WTD variation as an important driving variable, other responses can be focused on temperature effects or the lack thereof. We added the following text to the paper to reflect these observations (page 3 lines 1-7):

“During the DPH experiment, we measured water table depth in each plot (30-min measurement frequency), and did not observe—nor did we expect—any changes in water table elevation that were attributable to the deep peat warming treatment. Water tables were usually within 20 cm of the mean hollow surface and fluctuated over an approximately 40 cm range due to rainfall, snowmelt inputs, near-surface lateral flow, and evapotranspiration, but were not influenced by DPH. In addition, water table dynamics inside experimental plots mirrored those measured in ambient reference plots and the surrounding bog.”

2. Somewhat related to the above point - the term "peat surface" referred to in the manuscript is somewhat vague, presumably it means a layer between the top of the moss and some depth, is it the acrotelm or something else. A short discussion or clarification would help.

We have made a clarification on page 3 lines 14-17 that “surface” in this context refers to a depth increment 20-30 cm below the hollow surface. This layer is completely within the acrotelm, which extends to approximately 30 cm deep, and was inundated and anaerobic at the sampling times (Page 3 lines 14-17):

“Consistent with these field emission results, when peat was incubated anaerobically within 1°C of in situ temperatures, CH₄ production in surface peat (20-30 cm below the hollow surface) increased with temperature ($p < 0.001$) (Figure 2a). This layer is within the acrotelm¹⁴, but was consistently anaerobic at the time of sampling.”

3. In relation to Extended data Figure 1, the statement "In the absence of air warming during this phase of the experiment, anticipated energy loss at the surface resulted in less warming of the surface peat", which occurs in both figure caption and the manuscript body (pg. 2 lines 21-24) seems odd as the data mostly show warmer temperatures at the surface of the peat in this diagram. This is likely true in longer term average temperatures, but not in this figure.

We clarified the text to reflect that the +9°C separation among treatments that was achieved at depth was diminished at the surface due to energy loss (page 2 lines 21-24):

“The absence of air warming during this phase of the experiment resulted in heat loss at the surface, creating less separation among temperature treatment plots in the shallow peat relative to the 2 m depths (Extended Data Figure 2b).”

and revised the Figure caption as follows:

“Extended Data Figure 1: The seasonal progress of absolute peat temperatures at 2 m below the hollow surface throughout the DPH treatment period (a) and the temperature depth profiles associated with the June 16, 2015 coring event (b). This coring event took place 10 months after the deep peat temperature differentials were stable. In the absence of air warming during this phase of the experiment, anticipated energy loss at the surface reduced the separation among treatment temperatures.”

4. Some comment on the realism of this experimental design is needed. It is understandable why the experiment was set up the way it was, but i) warming of peat by climate change will not happen from below, heat must be transferred down the peat profile to the deeper

peats, and ii) the manipulations >+2 degrees C are likely far beyond what we can expect to occur at this depth in the peat with even the most liberal of climate change estimates for the next century. So, what does this experiment say about the real future?

Deep peat will naturally warm in parallel with surface warming due to propagation of heat downwards through the peat column—albeit at a small temporal offset. To quote from Hanson et al. (2011) “Often overlooked in warming studies is the reality that deep soil temperatures will also become elevated as they equilibrate with new mean annual temperatures (Baxter, 1997; Hu & Feng, 2003). Such deep warming has not been achieved using previous warming technologies. Instead, preferential heating of surface soils has occurred as soils deeper in the profile remain unaffected.” The SPRUCE study approach allows for such warming.

If warming experiments were infinitely large and long, air warming alone would drive deep peat warming. However, to achieve deep warming in a tractable timescale for experimentation on small plots that leak energy to the side, active heating of the peat at depth is required (Hanson et al. 2011). Heating is done in a very mild manner causing changes in deep temperatures that are not unlike the rate of change associated with changes over a full seasonal cycle. That is, to increase the deep peat temperatures to +9 °C took as much as three months.

Though it is hoped that +9°C is an upper limit on climate effects, the highest trajectories predict that temperatures in the Arctic may increase by as much as 8.3°C±1.9°C and extreme heat events are projected that may expose peatlands to acute heat stress exceeding conditions for which vegetation is currently adapted. Further, based on current greenhouse gas production rates (Raupach et al. 2007), forcing estimates greater than the A1B scenario are likely (Christensen et al. 2007). By increasing temperatures to +9°C above ambient, we can explore threshold response surfaces and the multiple temperature treatments allow non-linear curve fitting that may prove vital to understanding the response of ecosystems to warmer temperatures (Amthor et al. 2010). Therefore, we consider greater levels of warming than the ~4°C projected by the IPCC report. We have incorporated text to this effect on page 2 line 24 through page 3 line 1:

“Deep peat is expected to warm naturally, in parallel with surface warming, due to the propagation of heat downwards into the peat column. However, to achieve this effect in a tractable timescale for experimentation, active heating of the peat at depth is required⁷. While the highest climate trajectories project temperature increases up to +8.3°C (±1.9°C) in the Arctic between 2081 and 2100¹², the +9°C treatment employed in this study is an upper limit on what can be expected under the most extreme scenarios. We employ this treatment to explore threshold response surfaces to temperature change¹³ (e.g., Epping *et al.* ¹³) and because the multiple treatment effects above the median +2°C temperature projection allow for non-linear curve response fitting.”

Baxter DO (1997) A comparison of deep soil temperature: Tennessee versus other locations. Transactions of the ASAE 40, 727-738.

Hanson PJ, Childs KW, Wullschlegel SD, Riggs JS, Thomas WK, Todd DE, Warren JM (2011) A method for experimental heating of intact soil profiles for application to climate change experiments. *Global Change Biology* 17:1083–1096.

Hu Q, Feng S (2003) A daily soil temperature dataset and soil temperature climatology of the contiguous United States. *Journal of Applied Meteorology*, 42, 1139-1156.

Please also see our response to the first comment from Reviewer #3.

5. Minor point, but final line of the manuscript (pg. 6 lines 7-9) is a bit self-serving and likely not needed.

We have revised this final line as another reviewer pointed out that the longevity of the SPRUCE experiment may allow us to infer other relevant drivers of change (page 8 lines 8-11):

“The long-term SPRUCE experiment will enable us to examine whole-ecosystem warming, enhanced atmospheric CO₂ and water table feedbacks to these treatments, allowing us to clarify the internal mechanisms that control C cycling in a bog over a decade-long manipulative climate change study.”

Also, please see our response to the first comment from Reviewer #3.

Reviewer #2 (Remarks to the Author):

This study presents results on the fate of peat carbon stocks from a novel experiment in which peat was warmed in site to a depth of 2 m. The authors investigate the decomposition of the organic matter reservoir using a variety of methods including surface fluxes of CO₂ and CH₄, laboratory incubation of CO₂/CH₄ production, microbial community analysis, dissolved carbon pools and ages, etc. Overall, the various lines of evidence suggest that warming of the peat is unlikely to result in organic matter oxidation, but may shift the balance towards higher CH₄ flux. Given the large peat stock of C and potential for a feedback to climate change, the findings are of broad interest and deep warming of peat is novel.

There are a few inconsistencies in the data interpretation that need to be addressed prior to publication (e.g. statement of no clear temperature response for deeper peat in incubations, although there is clearly an increase from the figures

We believe the reviewer is referring to Figure 2b and suspect they have interpreted the figure incorrectly. Data are plotted relative to *in situ* temperatures—not treatment differentials—as noted in the figure caption. Also, as noted in the figure caption, there was a distinct seasonal trend in methane production that resulted in a bimodal response between sampling dates. The higher methane production in the earlier time period in the experiment results from seasonally warmer absolute temperatures in the summer, but does not reflect a response to the temperature treatment to which there is clearly no response.

and then this response varies compared to the field fluxes), but overall the methods and interpretations are sound. The one thing I would suggest to add is a short discussion on mechanisms for the protection of peat from increasing rates of decomposition under warming temperatures (the why?) even if the answer is still unclear. This will help to push the research forward by identifying what is still unknown.

Tfaily et al. (2014) showed a marked decrease in the o-alkyl content of catotelm peat relative to acrotelm peat at S1 bog, indicating intensive decomposition of carbohydrates. Recent studies have linked o-alkyl content to peat reactivity (Baldock et al. 1997; Leifeld et al. 2012) and have found clear decreases in o-alkyl peat content from northern peatlands to tropical peatlands. The tropical peatlands exist, with apparent stability in the face of elevated temperatures, similar to the catotelm peats in the SPRUCE experiment (Hodgkins, et al., in preparation). Thus, we hypothesize that the lack of reactivity of SPRUCE deep peat was due to low o-alkyl C content. We have added the following text in response to this comment (page 6 lines 23-page 7 line 3):

“While there is evidence of kinetic control on surface peat decomposition in our experiment, non-kinetic factors—such as chemical recalcitrance¹⁴—appear to be controlling the decomposition of deep C at S1 bog. Tfaily et al.¹⁴ report a marked decrease in the o-alkyl C content of catotelm peat relative to acrotelm peat at S1 bog, indicating intensive decomposition of carbohydrates. Previous studies have linked o-alkyl C content to peat reactivity^{28, 29} and have observed clear decreases in o-alkyl peat content from northern peatlands to tropical peatlands (S. Hodgkins, pers. comm., 2016).

Thus, we hypothesize that the lack of reactivity of SPRUCE deep peat was due to the low o-alkyl C content of the soil organic matter. Therefore, future warming will likely have little effect on the conversion of catotelm C to CO₂ and CH₄. However, catotelm peat recalcitrance is a relative term. We have shown that catotelm peat is recalcitrant with respect to temperature under its present conditions—water saturated, with fermentation and methanogenesis as the dominant organic matter decomposition processes.”

J. A. Baldock, J. M. Oades, P. N. Nelson, T. M. Skene, A. Golchin and P. Clarke. Assessing the extent of decomposition of natural organic materials using solid-state ¹³C NMR spectroscopy Aust. J. Soil Res., 1997, 35, 1061-1083.

Leifeld, J., M. Steffens, and A. Galego-Sala (2012), Sensitivity of peatland carbon loss to organic matter quality, Geophys. Res. Lett., 39, L14704, doi:10.1029/2012GL051856.

Finally, the authors have observed increases in CH₄ flux and a shift to lower CO₂:CH₄ ratio in the lab incubations. They make some statements at the end of the paper about the importance of this to GWP of peatland in response to deep warming, suggesting that even if the C stock is stable, peatlands may become a GHG source. I suggest that author should carefully consider how confident they are with this finding is and whether or not it should be included in the abstract (currently it is not). They end the paper with it, highlighting its importance, so I would suggest that it should be included in the abstract in this case.

We have added the indicated points to the abstract (page 1 lines 25-30):

“Higher CH₄ emissions, coupled with decreasing CO₂:CH₄ production ratios in surface peat are troubling given the radiative potency of CH₄. Under warmer conditions heightened plant biomass production and soil C sequestration may be unable to offset surface-driven CH₄ flux increases on this timescale, but our results suggest—while anoxia prevails—the large reservoir of deep peatland C will resist decomposition under future climatic warming.”

Specific comments:

Page 1, line 29: "rapid" can mean different things to different people. Hundreds of years can be considered rapid on a geological time scale. Can you be more precise here? Also, don't you think the results would warrant a strong statement here - possibly "changes to the large reservoir are very unlikely" in place of "may be minimal"?

We have amended the indicated text as shown in the previous response to remove the ambiguous term “rapid” and made the suggested change to strengthen the statement.

Page 2: Do you need the heading "Introduction". There are no other headings and the text here is actually introduction, results and discussion.

We removed the heading.

Page 2, line 1: As the deeper peat is relatively well insulated by the surface peat, how much warming is really expected at depth, particularly down to 2 m? I agree it will eventually warm in response to the warming atmosphere but won't this happen very slowly? Is the warming induced in the experiment representative of climate change?

Warming at depth in parallel with climate warming is expected, and actually happens quite rapidly (Amthor et al. 2010). Huang (2006) discusses the relationship between air and deep soil temperatures. Though it is hoped that +9°C is an upper limit, the highest trajectories predict that temperatures in the Arctic may increase by as much as 8.3°C±1.9°C and extreme heat events are projected that may expose peatlands to acute heat stress exceeding conditions for which vegetation is currently adapted. Further, based on current greenhouse gas production rates (Raupach et al. 2007), forcing estimates greater than the A1B scenario are possible (Christensen et al. 2007) and the high temperature treatments allow us to explore threshold responses and the multiple temperature treatments allow non-linear curve fitting that may prove vital to

understanding the response of ecosystems to warmer temperatures (Amthor et al. 2010). These points have been made explicit on page 2 line 24 through page 3 line 1:

“Deep peat is expected to warm naturally, in parallel with surface warming, due to the propagation of heat downwards into the peat column. However, to achieve this effect in a tractable timescale for experimentation, active heating of the peat at depth is required⁷. While the highest climate trajectories project temperature increases up to +8.3°C (±1.9°C) in the Arctic between 2081 and 2100¹², the +9°C treatment employed in this study is an upper limit on what can be expected under the most extreme scenarios. We employ this treatment to explore threshold response surfaces to temperature change¹³ (e.g., Epping *et al.* ¹³) and because the multiple treatment effects above the median +2°C temperature projection allow for non-linear curve response fitting.”

Huang S (2006) Land warming as part of global warming. *EOS*, 87(No. 44), 477, 480.

Page 2, line 4: Is reference to Hanson et al. 2011 really the best here. That study tests methods to heat deep soil, but does not determine that deep heating will occur in response surface heating

We have, at the indicated point, added references to Amthor et al. (2010) and Hu and Feng (2003) which do specifically address peat warming in theoretical terms and Huang (2006) which provides empirical evidence of the process.

Page 2, lines 22-23: So, without the surface heating the temperature profile is not really realistic due to the heat loss at the surface. Does this have implications for the interpretation of the results? Since incubations suggest the 20 cm depth was most impacted by the increase in temperature, the observed temperature profile may have limited the impact on surface fluxes compared to a profile warmed uniformly. Please comment.

Agreed, despite limited surface warming, we saw a response in the surface suggesting that added surface warming may increase the surficial response. We made this clearer by highlighting that this response was observed despite the “muted” surface temperature treatment (page 7 line 23-27):

“Further, these surface responses were underestimated due to energy loss at the surface that muted the warming treatment in surface peat. With surface warming, it is likely that the surficial response will be even greater. Thus, even if warming stimulates plant biomass production and enhances soil C sequestration it is unlikely these effects will completely offset the increases in CH₄ flux on this time scale.”

Page 2, lines 28-29: What happened to the plant community in response to heating? Plant respiration can dominate ecosystem respiration during the summertime, and changes in plant cover might mask differences in soil respiration.

Deep peat heating had no effect on plant cover over the duration of this study. While it is true that plant production and respiration is likely a major driver of surface processes in the field, the deep peat heating mostly affected below the active rooting zone for the plants at the site. Also, results from the incubations which did not include plants should control for the effects of net primary productivity on the soil response to heat treatment. Seasonal patterns of CO₂ efflux associated with plant activity are always present, and are implicitly involved in any temperature relationship-based data collected over time. The lack of a CO₂ response to deep heating during the growing season, in contrast with incubation results would indeed be difficult to verify from surface flux if the impacts of deep heating in surface soils are small (page 3 line 31 – page 4 line 4):

“...reflecting only the response of heterotrophic processes to temperature since photosynthetic and aerobic processes were excluded by the incubation design. These results differ from the field, where dark CO₂ flux did not correlate with temperature treatment, possibly because autotrophic processes were excluded in the incubations, or because CO₂ production in the field was greatest at depths shallower than 20 cm”

Page 3, lines 21-22: Do you think this is a fair statement? The relationship between CO₂ production and temperature was not significant, but there appears to be a clear increase after 13 months

Again, we think these statements are based on a misinterpretation of our figures. Please refer to introductory response to Review #2.

So what is protecting the peat from decomposition? Recalcitrance? Anoxia? Build-up of end-products? The incubations suggest you can release of this as CO₂ production does increase, but it doesn't happen in the field...

We disagree that the protection of deep peat from decomposition is relaxed in the incubations. The incubations showed a significant temperature effect, for both CO₂ and CH₄, only in incubations of surface peat. The difference between the surface peat incubations and the field is likely the exclusion of the aerobic peat layer and net primary productivity (page 3 lines 29- page 5 line4). The catotelm peat—which does appear to be protected from decomposition in both the field and the incubation studies—is resistant to anaerobic decomposition. The peat is apparently recalcitrant under the low energy yield of anoxic-methanogenic conditions. Introduction of oxygen to the peat would likely provide an electron acceptor of sufficient energy to allow the peat to decompose. It is also possible that other electron acceptors, such as sulfate or nitrate, would release the energetic constraints on decomposition of deeper peat. It is the absence of a sufficiently energetic electron acceptor that prevents the recalcitrant peat from decomposing. That is, the chemical conditions of the bog would have to be substantially altered for the peat to decompose. As stated above, the recalcitrance of the peat is likely a consequence of its low O-alkyl content (Tfaily et al., 2014; Baldock et al. 1997; Leifeld et al. 2012) and the conditions under which it is stored: water-saturated, anoxic, fermentative and methanogenic. We have added text describing this mechanism of protection to page 6 line 23 - 30.

Page 3, lines 23-24: Has anyone else observed this change in ratio with temperature? There have been a lot of peat incubations, so I'd be surprised that there is nothing you can cite here.

We incorporated references to Updegraff et al. (1996) and Yvon-Durocher et al. (2014) which show a similar decrease in CO₂:CH₄ ratios at higher temperatures.

Page 5, Lines 24-27: This is a pretty big however. If the deeper peat is exposed to different conditions due to water table drawdown or the oxidation of the surface layers and exposure, will it still respond in this way, or will the "protection" be gone. This is pretty critical for concluding that changes in the deep peat C will be minimal. I agree with you, but maybe it is not as clear cut.

Quantifying the relative magnitude of peatland response to increasing temperature versus hydrologic changes has been identified as a major uncertainty in determining the climate forcing impact of peatlands (Shädel et al. 2016). This study solely addresses increased warming, providing valuable data on the response to that forcing. See comments above.

Shädel, C., et al. Potential carbon emissions dominated by carbon dioxide from thawed permafrost soils. *Nature Climate Change* in press 1-5pp. (2016).

Page 9, lines 13-14: What do you mean here. The seasonal flux was fit to an exponential regression relating it to what?

We have revised the text to clarify that we regressed against temperature (page 12 lines 17-19):

“Seasonal flux measurements were fit against the average temperature from 1 to 2 m below the hollow surface with an exponential regression model using SigmaPlot v 12.3 and significant relationships identified at $p < 0.05$.”

Page 9, line 17: How were the intact cores collected and how large were they?

The Russian cores were 5 cm diameter, but the corer only collects approximately one half of a full cylinder. We have clarified the procedure in the text (page 12 lines 25-28):

“To prevent compression of surface peat samples, a serrated knife was used to collect a 10cm-diameter core from the hollow surface to approximately 20cm within the peat profile. A 5cm-diameter Russian corer was subsequently used to extract the remaining samples up to a 2m depth.”

Page 9, line 24: How long did the incubations last?

The incubations lasted ten days. We have revised the text to include this information (page 13 line 1-2):

“CO₂ and CH₄ production concentrations were determined during the course of the 10 day incubation.”

Page 10: lines 7-8: Were the piezometers sealed? Was there any exposure of the porewater at depth to oxygen? Clarify in text.

Clarified that the piezometers were covered, but not sealed when not being actively sampled from, that the inner diameter of the piezometer tubes was less than a cm so oxygen diffusion was limited and the tubes were flushed 24 hours prior to sampling so exposure to oxygen was minimized (page 13 lines 17-20):

“Piezometers were covered, but not sealed, when not being actively sampled, the diameter of the piezometers was less than 1 cm which limited oxygen diffusion, and piezometers tubes were pumped dry 24 hours prior to sampling to ensure that the sampled water was not in prolonged contact with the atmosphere prior to sampling.”

Page 11, line 3: There's an extra space here in the word "sample"

The extra space was deleted.

Extended data figure 2: For CO₂ flux, it almost looks like 2 relationships here with a break at around 9 degrees. Would you expect CO₂ flux to be linked to deep (1-2 m) soil temperature given that a lot of the source is the near surface oxic peat (and the plants in the summer)? Is there a relationship with surface peat temperature? And again, I think the CO₂ flux is less easy to interpret because it includes the plant respiration, which is likely variable between plots and might mask some of the patterns. Can you comment on how comparable the plant cover was between the various plots and different treatment temperatures?

In our opinion, interpreting the results in Extended Data Figure 2c as two relationships with a break at 9°C would be an over-interpretation as there are only two temperatures below 9°C. There is no relationship with surface peat temperature.

Plant cover varies only slightly among the plots measured for surface CO₂ and CH₄ efflux. All plots have a nearly uniform cover of Sphagnum over the hummock-hollow complex over which an ericaceous shrub layer is present. During the summer months, limited populations of the forb *Maianthemum* and some sedges also occupy the space. This has been noted in the “Site description” (page 10 line 42-page 11 line 10):

“Overstorey vegetation is dominated by two tree species, *Picea mariana* (black spruce) and *Larix laricina* (larch), while the understorey is composed mainly of low ericaceous shrubs, such as *Rhododendron groenlandicum* (Labrador tea) and *Chamaedaphne calyculata* (leatherleaf), as well as the herbaceous perennials *Maianthemum triflorum* (three-leaved Solomon’s seal) and *Eriophorum vaginatum* (cottongrass). The bog surface is characterized by hummock and hollow microtopography, with *Sphagnum magellanicum* colonizing the hummocks and *S. angustifolium*

the hollows. Typically, the hummocks are 10-30 cm higher than the hollows. Plant cover varies only slightly among the plots measured for surface CO₂ and CH₄ efflux. All plots have a nearly uniform cover of Sphagnum over the hummock-hollow complex over which an ericaceous shrub layer is present. During the summer months, limited populations of the forb *Maianthemum* and some sedges also occupy the plots.”

Extended data Figure 5: There is definitely higher CO₂ production after warming at all depths, but then no legacy effect. What do you think the cause is? Is this a more active microbial community or a change in substrate quality or both? Why is it not reflected in the surface flux?

This is incorrect. CO₂ production is *lower* in incubations of peat collected after 13 months of DPH—depicted by the open symbols in Extended data Figure 5 b. The closed symbols represent CO₂ production from peat collected 4 months after initiation of DPH. Moreover, at overlapping temperatures, rates from the later sampling event were consistently lower. It is not clear what caused this seasonality in the deep peat CO₂ production, but it is not a simple temperature effect. Thus, there is a consistent lack of a temperature effect on CO₂ production in deep peat and in surface fluxes.

However, it is true that we observed a positive temperature response for both CH₄ and CO₂ production in the surface, but no legacy effect. This implies that the increase was kinetically-driven (enhanced microbial activity) and that DPH did not generate any lasting effects on the microbial community structure or the peat itself (substrate supply and quality). While we do observe a similar response in CH₄ flux, it is likely this is not reflected in CO₂ flux because of the presence of the plant community and/or the inclusion of aerobic processes.

Reviewer #3 (Remarks to the Author):

As the apparent initial phase of a long-term ecosystem manipulation project, Wilson et al., investigated how CO₂ and CH₄ fluxes varied following 1 year of subterranean (~ 2 m) heating (0.0oC to 9oC above ambient) of a peatland. Through surface level measurements of CO₂ and CH₄ evolution at plots subjected to varying degrees of warming the authors observed that CH₄ fluxes from the surface increased. However, based on in situ measurements of community composition, enzymatic activity, and isotopic enrichment analyses along with laboratory incubations of peat from the surface and the deeper biosphere they determined that this flux was due to turnover of peat at the surface and not the deeper (> 1 m) carbon pool.

Overall, this is a well written and presented study addressing a very important questions. Specifically, since peatlands contain a substantial portion of terrestrial carbon therefore it is important to predict how the fate of this carbon might be affected due to climate change. However, there are two major criticisms that cause this manuscript to fall short of demonstrating what this statement at the end of the abstract "Our results indicate that rapid changes to the large reservoir of deep buried C in peatlands may be minimal under future climatic warming."

Major concerns:

1-- the experimental design used here seems to limit the relevance of this data to the larger ecosystem. In particular, the authors do not establish the validity of heating below ground without corresponding manipulations to the surface. This is important since I am unaware of any global warming models where heating originates from the deep subsurface and radiates upward. By conducting the ecosystem manipulation in this manner, they effectively limit the impact of their study as it poorly accounts for potential, if not likely, changes to the ecosystem that may originate at the surface which could have cascading effects that reach the deeper parts of the soil column. The authors mention this to a degree (pg.

5 ln 27 - 31), but it doesn't seem to be taken into account in the experimental design. Thus, the results from this study seem to be more of a baseline observation on the limiting effects temperature alone has on the deep carbon pool, if nothing else is changed. This is certainly of some interest given the importance of the problem, however, the relevance to native ecosystem processes would need to be established to have broad appeal to the scientific community.

Surface warming is a critical aspect of the whole-ecosystem response and will be investigated in the next scheduled phase of the experiment. In this initial phase of the manipulation, our objective was to investigate the effects of temperature increase on the decomposition of deep C pools in the absence of other cascading effects that might serve to confound interpretation of results once surface heating was implemented. Our approach was to isolate one variable at a time. We prefer to view the deep peat heat treatment as the first phase of the manipulative study rather than a baseline. We collected several years of baseline data prior to the initiation of the treatments to fully characterize the conditions across the site. The data presented in this paper regarding the response to deep peat heating is of significant interest because it shows that under the current water-saturated methanogenic conditions, the deep catotelm peat carbon store is resistant to decomposition under warmer conditions indicating that the effect is not kinetically limited—i.e. warming alone will not result in the wholesale conversion of catotelm C to greenhouse gases. It seems likely that the recalcitrant catotelm C decomposition is thermodynamically limited by the absence of suitable electron acceptors. Subsequent cascading effects that provide more energetically favorable electron acceptors, such as hydrological changes that increase O₂ availability, could increase catotelm peat decomposition. However, we show in this paper that temperature effects alone will not have that result. The response of peatlands to global warming is a multi-faceted problem with both additive and opposing effects, understanding the details of each of major effect—such as deep peat heating—will improve modeling inputs providing better predictive capabilities moving forward.

2--The authors conducted several experiments to both localize methane production in the column and assess possible reasons why deep carbon wasn't affected, with consistent results: temperature did not appear to affect microbial composition, enzymatic activity or greenhouse gas production from the deeper soil column. However, a second major criticism is that in spite of the multiple incubations and measurements conducted, the study lacks a few experiments that are needed to help solidified the cause of this persistence and thus strengthened this study. Specifically, is the deep peat inherently more resistant to microbial decomposition or is this persistence due to the microbial composition or the environmental conditions deeper in the profile (e.g. low pH, oxygen limitation)?

We hypothesize that the peat recalcitrance is related to a decrease in o-alkyl carbon (Tfaily et al., 2014) as these parameters have been shown to be linked in previous studies ((Baldock et al. 1997; Leifeld et al. 2012) and acknowledge that this persistence is maintained by specific environmental conditions at depth as noted above.

If it is actually abiotic/biotic conditions deeper in the profile that are the controls, not the peat itself, prolonged surface warming may alter the abiotic/biotic factors. While they do mention this to some degree on page 5 line 23, additional incubation experiments such as incubation of this older peat with populations or conditions more similar to the surface environment would be required to establish that it is the properties of the peat that limit decomposition.

We disagree that transplanting the microbial community from the surface to incubations of deep peat would provide a realistic assessment of whether the microbial community itself is limiting deep peat decomposition. Rather, if the microbial community was the medium of indirect temperature effects, we would expect to see a shift in the microbial community in the heat treated incubations. In fact, we saw no changes in the microbial community with temperature treatment. The surface microbial community appeared to be correlated with oxygen availability indicating that the availability of energetic electron acceptors is the limiting factor. Warming the surface peat did not significantly alter the microbial community nor did warming the

deep peat make the microbial community more similar to the surface. In terms of the *effects of warming on deep peat*, we have rigorously tested the response of the microbial community and determined that the microbial community does not exhibit a significant response to warming.

This likely will reveal, as the authors suggest as a possibility, that oxygen concentration is limiting deep peat mineralization, which would be an important finding. However, if this deep peat is truly resistant to microbial decomposition as a result of its chemical structure and not other biotic or abiotic factors, this would also be of high interest.

The reviewer makes an interesting point. Recalcitrance is a relative term. We have shown that the catotelm peat is recalcitrant with respect to temperature under its present conditions – water saturated, with fermentation and methanogenesis as the dominant organic matter decomposition process. We agree that our results do not allow prediction of the decomposition response were oxygen to penetrate into the catotelm. Under those conditions, the peat would likely not be recalcitrant. Added page 7, lines 1 to 17:

“However, catotelm peat recalcitrance is a relative term. We have shown that catotelm peat is recalcitrant with respect to temperature under its present conditions—water saturated, with fermentation and methanogenesis as the dominant organic matter decomposition processes. Other climate-induced perturbations to the ecosystem—changes in water-table depth, increased plant productivity, belowground exudation of labile plant compounds, or changes in plant communities—could have cascading effects on peatland C dynamics. For example, lowering of the water table due to increased evapotranspiration could increase O₂ availability providing the necessary conditions for degradation of recalcitrant phenolic compounds in the catotelm, which have been proposed to protect the global C bank in deep peat through inhibition of microbial heterotrophy according to the “enzyme latch” hypothesis³⁰. However, recent studies have shown that temperature, water-table depth, and perhaps even nutrient availability may control the strength of the enzyme latch and that therefore the response of phenolic compound degradation to climate drivers may be more complicated than originally hypothesized^{31,32}. While peat decomposition was enhanced in incubation of surface peat, our results provide evidence that C decomposition in deep anaerobic peat is not kinetically constrained; therefore, peat decomposition is most likely thermodynamically limited by the absence of suitable electron acceptors.”

Minor points:

- The exponential regression in figure 1a looks to be a poor fit to the data ($r^2 < 0.14$ in both cases) and therefore it is unclear why this is shown or described as an exponential relationship (pg 2., lines 25 - 26).

Enzymatic responses to temperature are theoretically considered to be exponential, as modeled by an Arrhenius equation. The regression explains a small but significant amount of the variability in the data. Because the regression fell within our predefined criteria of significance ($p < 0.05$), we presented the results.

- Please be consistent with labeling figure and extended figure captions in the descriptions. In some cases the label (i.e., 'a','b','c') is at the beginning of the description and at other times it is at the end.

The figure captions have been revised to introduce labels prior to introducing the panels.

- Pg 4, line 22 - what is the difference between the 'depth' and 'magnitude' of a shift?

We have clarified the text to indicate that we mean the depth at which the shift occurred and the magnitude of the isotopic change (page 5 lines 13-14):

“The magnitude of the isotopic shift as well as the depth at which the shift occurred was similar across temperature treatments (Figure 3b)...”

- Pg 3, last paragraph - if there were not consistent effects observed due to temperature changes, why would you expect to see 'legacy effects'?

The reviewer raises an interesting point. In situations where there were no consistent effects of temperature on CO₂ and CH₄ production (i.e., the lack of warming impact on CH₄ production in deep soils), we would not expect legacy effects. However, in cases where temperature did have an effect, incubation at a common temperature helps to evaluate potential legacy effects. This is most important in the 20-30 cm depth where incubations show an increase in CH₄ production (Figure 2a). The lack of legacy effect of 15 months of heating at this depth (Extended Figure 7) suggests that the temperature response of CH₄ was not driven by changes in soil carbon quality of CH₄ production potential in these soils.

- A schematic drawing of the site would be helpful in the SI.

Agreed. We have added the following schematic to the supplemental information—

- Extended data, figure 8. Why was only the top 2 cm examined?

There was an error in the vertical axis label. That should be 2 meters, not 2 centimeters. This has been corrected in the current version of the figure.

4. Review of Wilson et al. "Stability of 1 peatland carbon to rising temperatures"

I think the main finding of the non-responsiveness of deeper peat to even dramatically increased peat temperature is a very important finding. This finding, if it can be universally applied, is very good news for the planet: it means that a significant component of what made up the proposed northern "carbon bomb" is not a "bomb" but more like a fire cracker, and a pretty small one at that. This finding is well supported by the several lines of evidence the authors present. I am very impressed with how tight the authors' arguments are and the varying lines of evidence they bring forward to support the case. I would say they have pretty much nailed it on this part of the manuscript. They are also careful to differentiate their findings

from those of others who have worked on deeper peats in permafrost peatlands. I do think it is important for the authors to add a sentence to emphasize their results apply to ombrotrophic bogs and do not necessarily apply to transitional or blanket bogs, or mineral poor to mineral rich fens. Their results may apply in some cases but they provide no evidence or argument as to why they should apply. Why this is important is a significant amount of carbon stored in peatlands is stored in other systems than bogs. For example, if we look at the Canadian peatland statistics about 70% of peatlands are bogs and this represents about 130 Gt C in Canada alone, but the remaining peatlands are mostly fens (~30 Gt C in Canada).

We have included a statement highlighting that our findings are specifically applicable to ombrotrophic bogs and may not be generalizable to fens or permafrost environments (page 6 lines 20-23).

“It should be noted that the lack of response reported here may be specific to ombrotrophic bogs and does not necessarily reflect the expected or observed response from peatland habitats such as fens or permafrost peatlands.”

It should be noted that S1 bog chemistry indicates that at depth acidity decreases, nutrients increase, and ion concentrations increase suggesting that there is a fen beneath the bog—a feature common in blanket bogs. In this way S1 bog is very representative of a vast number of bogs (including blanket and transitional bogs) that exhibit ombrotrophy at the surface, but minerotrophy at depth.

I am less impressed with the significance of the finding of increased methane emissions from the near surface peat, or the lack of response in HR. The authors results are interesting, but I am concerned about the treatment design too put much as much stock in these results. Firstly, it is a manipulation experiment and the authors have subjected their peatlands plots to a very large and relatively sudden temperature increase. From the results of other manipulation experiments we know there can be a very large initial change and then over time the change becomes less. The transient nature of the response of ecosystems early in the treatments make them very hard to interpret. Many of the authors on this manuscript have done other manipulation experiments and it is their papers that show the initial large response and then a subsequent lesser response. I think the authors need to acknowledge at minimum what happens to ecosystems when they are pushed rapidly far from equilibrium.

We agree that some of the response that we see in the surface peat may be a transient perturbation effect such as has been observed in previous manipulation experiments. We have included text and relevant references addressing this issue on page 7 line 27- page 8 line 11:

“However we must temper our interpretation because the observed surface response may be a transient perturbation effect as has been seen in other climate manipulation experiments^{34, 35, 36}. In addition, increased frequency and duration of low water table elevations and flow along near-surface lateral flowpaths are most likely to affect surface peat, which may exacerbate or mitigate the responses that we observed^{37, 38}. For example, even with a temperature increase, a lowered water table could reduce CH₄ production, enhance oxidation and result in lowered CH₄ emissions. In peatlands feedbacks exist among plant communities, water table dynamics, and physical properties of the peat resulting in a tight coupling between C and water cycling^{34, 39} that allows the system to self-regulate, resisting gradual environmental change until a catastrophic tipping point is reached and the system shifts towards a new steady-state^{13,40}. For example, Sphagnum and vascular plants, respectively, alter environmental conditions such as light and nutrient availability, water table depth, temperature, and pH^{13, 34}. The long-term SPRUCE experiment will enable us to examine whole-ecosystem warming, enhanced atmospheric CO₂ and water table feedbacks to these treatments, allowing us to clarify the internal mechanisms that control C cycling in a bog over a decade-long manipulative climate change study.”

The second concern is that of hydrology. I have discussed the design of SPRUCE with several of the authors so they will not be surprised with this concern. The results of the deeper peats is not influenced significantly by the hydrology but the surface peats are – i.e. there is no way the water balance and energy balance of a bogs (in my experience through both measurements and modelling) will change the moisture contents much below a metre depth. However, I would expect the moisture contents to change significantly in the surface peats with temperature changes in the order used in this study. However, the experiment

does not explicitly experiment with changes in moisture content or lowering of a water table. If the authors look at the literature on fluxes of CO₂ and CH₄ from bogs they are quite responsive to variations in temperature, water table depths and/or soil moisture in the acrotelm. Later in the paper the authors acknowledge this limitation but this could come much earlier as I think it should temper and qualify the evidence presented on the response of near surface peat. For example, I would anticipate with such temperature increases there would have to be a commensurate decrease in acrotelm SMC and a corresponding increase in CH₄ oxidation. Are the authors results only providing part of the story (temperature response) while constraining another part of the story (moisture)?

We agree with the reviewer that peatland ecosystem processes are critically tied to changes in water table depth (WTD). During the deep peat heating experiment, described in this paper, we did not expect to see any change in water table dynamics because warming only targeted the deep, permanently saturated catotelm, nor did we intend to induce any unnatural changes that would have complicated the experimental design. During the DPH experiment, we measured WTD in each plot (30-min measurement frequency), and did not observe any change that was attributable to the deep peat warming treatment. Water tables fluctuated over an approximately ~40 cm range. Water tables were usually within 20 cm of the mean hollow surface and fluctuated due to rainfall, snowmelt inputs, near-surface lateral flow, and evapotranspiration, but with no apparent effect of deep peat heating. In addition, water table dynamics inside experimental plots mirrored those measured in ambient reference plots.

Though not relevant to this manuscript, we can address the reviewers comment with regards to our plans for the next phase of SPRUCE. Once aboveground warming is initiated and whole-ecosystem warming begins, we predict that WTD will change with temperature due to changes in precipitation form (rain vs. accumulation in a persistent snowpack), snowmelt timing, increased evapotranspiration, and natural outflow along near-surface flowpaths. In the SPRUCE experiment, WTD is considered a *response* variable in that water table will fluctuate in response to temperature using unique project infrastructure that allows passive, natural drainage along near-surface pathways.

Effects would not be limited to just soil moisture. Though the typical annual range in water table fluctuations is about 30 cm, long-term data from the S1-Bog and other peatlands in the research program at the Marcell Experimental show that fluctuations in water levels have ranged up to 1.4 m depth (Sebestyen et al. 2011) over the course of a 50-year record. The lowest water table elevations occurred during the worst drought on record. While the whole-ecosystem warming experiment may not include such a dry year, the experiment is expected to increase the partitioning of water to evapotranspiration, which may result in similar or even greater water table drawdowns and exposure of peats to oxygen during seasonal or prolonged periods. The interactive effect of >evapotranspiration and an equivalent severe drought may lead to unprecedented drawdowns in water table elevations.

Please also see the response to the first comment from reviewer 1.

I am not arguing the authors' results are not interesting, they are but they confirm the results of many previous laboratory studies. Those same laboratory incubations (again some done by the authors of this paper) also show the important of changes in moisture. I suggest the authors need to show more caution in the interpretation and significance of the changes, or lack thereof in the shallow peat. I think it is critical that they be much more cautious on the universality of those results and much more self-critical on the design of their treatments for assessing the responses as representative of what one would expect in natural systems. I think this can be handled very easily – emphasize the deep peat results and de-emphasize the near surface peat results, and add in some qualifications.

Our intention was to emphasize the lack of response in the *deep peat* to heat treatment. We have added text that explicitly acknowledges ecosystem feedbacks, such as changes in plant dynamics, that could dramatically affect surface peat processes (page 8 lines 2-11):

“In peatlands feedbacks exist among plant communities, water table dynamics, and physical properties of the peat resulting in a tight coupling between C and water cycling^{34, 39} that allows the system to self-regulate, resisting gradual environmental change until a catastrophic tipping point is reached and the system shifts towards a new steady-state^{13,40}. For example, Sphagnum and vascular plants, respectively, alter environmental conditions such as light and nutrient availability, water table depth, temperature, and pH^{13, 34}. The long-term SPRUCE experiment will enable us to examine whole-ecosystem warming, enhanced atmospheric CO₂ and water table feedbacks to these treatments, allowing us to clarify the internal mechanisms that control C cycling in a bog over a decade-long manipulative climate change study.”

My comments below related to specific parts of the manuscript but largely reflect the general comments above. I definitely think the lack of response of deeper peat is worthy of publication in Nature.

Pg 1 Ln 29 A critical finding and one that is global significant. The only concern is will this relative insignificant response continue. I am always concerned about short term responses from manipulation experiments.

We have made in-text amendments to acknowledge the length of our study (page 6 lines 19-20):

“Deep peat heating up to 9°C above ambient failed to stimulate catotelm C decomposition in this ombrotrophic bog within the first 13 months of this experiment.”

Similar statements are also made on page 4 lines 14-16, page 4 line 30 – page 5 line 1, page 6 lines 7-9, and page 6 lines 12-14. Additionally, the majority of our figure captions denote the length of our study and sampling times. We have also raised concern about the transient perturbation effects often observed in climate manipulation studies:

“However we must temper our interpretation because the observed surface response may be a transient perturbation effect as has been seen in other climate manipulation experiments^{35, 36, 37}.”

35. Bridgman, S. D., J. Pastor, *et al.* Rapid carbon response of peatlands to climate change. *Ecology*. 89, 3041-3048 (2008).
36. Luo, Y., Wan, S., Hui, D. & Wallace, L.L. Acclimatization of soil respiration to warming in a tall grass prairie. *Nature*. 413, 622-625 (2001).
37. Melillo, J.M., Steudler, P.A., Aber, J.D., Newkirk, K., Lux, H., Bowles, F.P., Catricala, C., Magill, A., Ahrens, T., & Morrisseau, S. Soil warming and carbon-cycle feedbacks to the climate system. *Science*. 298(5601), 2173-2176 (2002).

Pg 2 Ln 14 An important constraint that the SPRUCE team is well aware of is that deep warming is likely to occur with significant changes in moisture contents - i.e. peatlands are likely to become drier at the same time as they become warmer. While SPRUCE is not designed to test the effects of reduced WTDs and reduction in water contents in the unsaturated zone, this constraint should be recognized in the interpretation of the authors' results. What would be the result if the mean WTDs were to be reduced say 10 to 30 cm and the SMC were to decrease in the unsaturated zone by several ten percent?

We added to page 7, lines 29-page 8 line 2.

“In addition, increased frequency and duration of low water table elevations and flow along near-surface lateral flowpaths are most likely to affect surface peat, which may exacerbate or mitigate the responses that we observed^{37, 38}. For example, even with a temperature increase, a lowered water table could reduce CH₄ production, enhance oxidation and result in lowered CH₄ emissions.”

See also responses to above and below comments regarding the expected changes in water table level with whole-ecosystem warming.

The results for the surface peat is what I would expect based on numerous incubations but the in situ measurements are an important addition. However, the incubation experiments that have also included moisture change show the results from SPRUCE are different. The results for the deeper peats in not what I expected though are within the realm of possible results I would have anticipated. This is where the authors' results are unique and I think substantive in nature. First they show little response, which does vary with the other results reported in the literature, but more importantly the authors can convincingly explain what they got the results they did. That is, they provide the process level data to substantiate the system level response – or more correctly the lack of response of the deeper peat.

We appreciate the reviewer's comments.

Pg 3 Ln 7 This makes the issue of moisture change even more important to consider since it will be in the surface layers where the response seems to occur. From continuous EC measurements of CH₄ from bogs like Marcell, there tends to be a relationship with moisture contents albeit a complicated one (e.g. see Brown et al. 2014 but there are many more).

We have added text to indicate that water table dynamics did not respond to DPH and there was no expectation or intention that they would. Please see previous response(s) to comments on WTD changes above.

Brown, M. G., E. R. Humphreys, T. R. Moore, N. T. Roulet and P. M. Lafleur (2014). "Evidence for a nonmonotonic relationship between ecosystem-scale peatland methane emissions and water table depth." *Journal of Geophysical Research: Biogeosciences* 119(5): 2013JG002576.

Pg 4 Ln 15 Does the warming increase photosynthesis - i.e. the supply of exudates, and/or does the warming alter the production of more labile components of the exudates? What is the sources of the exudates?

We have added text to indicate that increased primary production would increase the production and availability of root exudates for decomposition (page 4 lines 28-30). Likely the next phase of the manipulation, the surface warming and surface warming with elevated CO₂ will better answer the reviewers question about the effect of warming on exudates and primary production (page 4 lines 28-30):

“Increasing temperatures are likely to stimulate photosynthesis rates and increase root exudation of organic C available for decomposition¹⁷.”

Pg 4 Ln 27-28 An important contrast! Could it be that the deeper peat in the permafrost setting is nowhere as recalcitrant as it is in Marcell? After all the peat is locked into the permafrost at ~ 50 to 70 cm. The work by Zimov and others on yedema, which I know is very different, has shown the C in permafrost when thawed supports a consider amount of heterotrophic respiration.

Yes, we believe that in permafrost settings the peat is frozen at a partially decomposed state, preserving labile material until such time as the peatland thaws and it can continue to decompose. In contrast, non-permafrost peatlands only experience freezing in surface peat during winter and spring. This apparently leads to greater decomposition and therefore upon heating, there is not much left to stimulate decomposition. We have added these points to page 5 lines 20-27.

“In particular, our results contrast those of permafrost peatlands exposed to thaw. In permafrost settings—particularly syngenetic permafrost—the organic matter is frozen at a partially decomposed state, that is, decomposition is suspended preserving labile material. As permafrost thaws, that labile material becomes available enhancing decomposition rates. In contrast, non - permafrost peatlands only experience seasonal freezing in surface peat, leading to millennia of slow decomposition of deep peat. In the case of S1 bog over the at least the time frame of this study, the result is temperature insensitivity of the decomposition of recalcitrant deep peat.”

Pg 5 Ln 15-17 The reason I raise the moisture issue.

Please see previous responses to comments on how water table depth varied during the study period.

Pg 5 Ln 21-23 This is the most important finding! The surface responses are interesting but may be highly transient. I think it would be a stretch to argue (I don't think the authors are arguing this) that we should expect continued elevated CH₄ from the surface layers or a lack of response of CO₂. This response may be very short term? The SPRUCE experiment should be able to answer this as long as the structural design of the SPRUCE does not inhibit the self-regulation between NPP and decomposition that exists in bogs. This is a longer regulation than the duration of SPRUCE. The authors should recognize that they are dealing with self-regulating ecosystems and this will have a profound effect on the difference between short-term and long-term responses. In addition, the design of SPRUCE may actually interfere with the ability of the ecosystem to self-regulate. I would have to think this through and possibly run some simulations with a model such as HPM (Frolking et al. 2010) or DigiBog (Morris et al. 2012, 2013, 2015) to see what the impact of restricting lateral water flow. Lateral water flow is a critical element to the self-regulation of peatlands (see Belyea & Baird 2006 or papers by Eppinga et al. 2009). I am not arguing the authors work on self-regulation but I do think they need to acknowledge that the ecosystem they are working on has this relative unique characteristics compared to most other ecosystems. In very wet and dry ecosystems (arid lands) water plays a far more critical role than it does in other ecosystems.

We appreciate the thoughtful comments of the reviewer, and we have attempted to incorporate these self-design elements as they relate to hydrology in the future SPRUCE design. Though not operated during DPH, the Whole-Ecosystem-Warming phase of SPRUCE includes infrastructure (a belowground corral) that prevents inflow of water during times when water tables inside plots relative to the surrounding bog would be experimentally lowered by enhanced evapotranspiration. In addition, the corral allows natural drainage along the existing shallow subsurface flowpaths when water tables rise into the acrotelm and the measurement of that natural outflow. The corral was installed during DPH, but natural, passive drainage did not start until after DPH ended.

These points have been addressed in the new text added on page 8 lines 2-11.

"In peatlands feedbacks exist among plant communities, water table dynamics, and physical properties of the peat resulting in a tight coupling between C and water cycling^{34, 39} that allows the system to self-regulate, resisting gradual environmental change until a catastrophic tipping point is reached and the system shifts towards a new steady-state¹³. For example, Sphagnum and vascular plants, respectively, alter environmental conditions such as light and nutrient availability, water table depth, temperature, and pH^{13, 34}. The long-term SPRUCE experiment will enable us to examine whole-ecosystem warming, enhanced atmospheric CO₂ and water table feedbacks to these treatments, allowing us to clarify the internal mechanisms that control C cycling in a bog over a decade-long manipulative climate change study."

In terms of the self-regulation of the peatland, generally bogs are domed and lateral water flow is from the center to the edge (lagg) so, theoretically, we are not restricting lateral flow into the chambers; however, there may be some inhibition depending on each chamber position within the bog.

- Belyea, L. R. and A. J. Baird (2006). "Beyond "the limits to peat bog growth": cross-scale feedback in peatland development." *Ecological Monographs* 76: 299-322.
- Eppinga, M. B., P. C. De Ruiter, M. J. Wassen and M. Rietkerk (2009). "Nutrients and hydrology indicate the driving mechanisms of peatland surface patterning." *American Naturalist* 173(6): 803-818.
- Eppinga, M. B., M. Rietkerk, M. J. Wassen and P. C. De Ruiter (2009). "Linking habitat modification to catastrophic shifts and vegetation patterns in bogs." *Plant Ecology* 200(1): 53-68.
- Frolking, S., N. T. Roulet, E. Tuittila, J. L. Bubier, A. Quillet, J. Talbot and P. J. H. Richard (2010). "A new model of Holocene peatland net primary production, decomposition, water balance, and peat accumulation." *Earth Syst. Dynam. Discuss* 1: 115-167.
- Morris, P. J., A. J. Baird and L. R. Belyea (2012). "The DigiBog peatland development model 2: Ecohydrological simulations in 2D." *Ecohydrology* 5(3): 256-268.
- Morris, P. J., A. J. Baird and L. R. Belyea (2013). "The role of hydrological transience in peatland pattern formation." *Earth Surf. Dynam.* 1(1): 29-43.
- Morris, P. J., A. J. Baird, D. M. Young and G. T. Swindles (2015). "Untangling climate signals from autogenic changes in long-term peatland development." *Geophysical Research Letters* 42(24): 10788-10797.

Pg 5 Ln 27-31 I do not like promoting our own work in reviews but there are a series of papers by Andrew Pinsonneault et al. that have just come out that the authors might want to look at. Chris Evans was Drew's external examiner (Evans is a coworker of Freeman's - the author of the ELH) and he thinks there is more to it than simply the ELH.

Pinsonneault, A. J., T. R. Moore and N. T. Roulet (2016). "Temperature the dominant control on the enzyme-latch across a range of temperate peatland types." *Soil Biology and Biochemistry* 97: 121-130.
Pinsonneault, A. J., T. R. Moore, N. T. Roulet and J. F. Lapierre (2016). "Biodegradability of Vegetation-Derived Dissolved Organic Carbon in a Cool Temperate Ombrotrophic Bog." *Ecosystems*: 1-14.

We have incorporated a reference to the Pinsonneault et al. articles acknowledging that the “enzyme-latch hypothesis” may be more complicated than originally indicated (page 7 lines 6-17).

“For example, lowering of the water table due to increased evapotranspiration could increase O₂ availability providing the necessary conditions for degradation of recalcitrant phenolic compounds in the catotelm, which have been proposed to protect the global C bank in deep peat through inhibition of microbial heterotrophy according to the “enzyme latch” hypothesis³⁰. However, recent studies have shown that temperature, water-table depth, and perhaps even nutrient availability may control the strength of the enzyme latch and that therefore the response of phenolic compound degradation to climate drivers may be more complicated than originally hypothesized^{31,32}. While peat decomposition was enhanced in incubation of surface peat, our results provide evidence that C decomposition in deep anaerobic peat is not kinetically constrained; therefore, peat decomposition is most likely thermodynamically limited by the absence of suitable electron acceptors.”

Pg 6 Ln 1-3 This is where I am concerned because it is where water is critical. For example the net ecosystem carbon balance of MB, the longest record for a bog (now 18 years) is correlated with mean growing season WTD as well as growing season temperature. We have presented this at conferences but the manuscript is still in preparation.

Please see above responses to concerns about the water table. There was no expectation that DPH would induce changes to water table elevation and indeed all variation in water table during DPH was consistent with long term records of seasonal variation at the site. From this we infer that we were successful in isolating deep peat heating as the sole treatment under investigation in this study.

Pg 6 Ln 5-7 This is again where the changes in moisture become critical.

We agree that water table will likely have a profound effect on surface peat decomposition. However, under DPH treatments and with the weather conditions from June 2014 through August 2015, fluctuations in water tables in deep heat treatment plots mirrored water table dynamics in the surrounding bog. Water levels rose and fell in response to rainfall and snowmelt and evapotranspiration. The typical annual range of water tables at five long-studied bogs on the Marcell Experimental Forest is 30 cm (Sebestyen et al. 2011). The ranges inside DPH experimental plots were similar and timings of changes inside DPH experimental plots were similar to the ambient locations outside the experimental plots. In the absence of WTD variation beyond typical natural variation within a year, these data do not allow us to speculate on the final fate of CO₂ and CH₄ emissions for DPH. We are, however, sustaining our observations of flux and WTD under whole-ecosystem warming to specifically track their relationship and the nature of gas efflux when greater depths of the peat profile become aerobic with warming.

REVIEWERS' COMMENTS:

Reviewer #1 (Remarks to the Author):

Congratulations I feel you have done an excellent job addressing the all of the reviewer's comments. Parts of this study will still be controversial to some readers, but that is often expected of good science.

Reviewer #2 (Remarks to the Author):

As in my original review, I believe that the authors present results from a novel study of deep peat warming in which they report on the limited impact of the warming on release of CO₂ from deep peat.

In their response to my own and the other reviewer comments, they have clarified important aspects of the study, added some qualifying statements that highlight the importance of hydrology and anoxic conditions for protecting deep peat. These changes have cleared up my previous concerns and I recommend publication.

Reviewer #3 (Remarks to the Author):

I am satisfied with the revisions and authors' responses. This is an important finding and will be of broad interest.

Reviewer #4 (Remarks to the Author):

I finally have had time to read the revisions and the response of the authors to all the reviewers' comments. First to my comments - the authors have addressed all my comments by revisions that clarify or make the connections that I thought were missing. They have softened some of their original statements. Secondly, I think they have done an excellent job at addressing my fellow reviewers' comments. As with mine they have taken each point seriously and have dealt with them.

I think this is a better paper. I think the results are important. I say this even though this manuscript if published will not help my personal search for research funding. However, these authors provide a very convincing argument with ample evidence to support their conclusions.

Now I want to know what happens to the other 30 to 40% of northern Peatlands that are mineral poor to rich fens. The authors now clearly state that their results do not apply to fens. Our work suggests these may be much more sensitive to change. There is a group in Canada that is now attempting a similar experiment on fens and we need to convince some of these authors to bring their techniques a little further north.

Reviewer #1 (Remarks to the Author):

Congratulations I feel you have done an excellent job addressing the all of the reviewer's comments. Parts of this study will still be controversial to some readers, but that is often expected of good science.

Thank you.

Reviewer #2 (Remarks to the Author):

As in my original review, I believe that the authors present results from a novel study of deep peat warming in which they report on the limited impact of the warming on release of CO₂ from deep peat.

In their response to my own and the other reviewer comments, they have clarified important aspects of the study, added some qualifying statements that highlight the importance of hydrology and anoxic conditions for protecting deep peat. These changes have cleared up my previous concerns and I recommend publication.

Thank you.

Reviewer #3 (Remarks to the Author):

I am satisfied with the revisions and authors' responses. This is an important finding and will be of broad interest.

Thank you.

Reviewer #4 (Remarks to the Author):

I finally have had time to read the revisions and the response of the authors to all the reviewers' comments. First to my comments - the authors have addressed all my comments by revisions that clarify or make the connections that I thought were missing. They have softened some of their original statements. Secondly, I think they have done an excellent job at addressing my fellow reviewers' comments. As with mine they have taken each point seriously and have dealt with them.

I think this is a better paper. I think the results are important. I say this even though this manuscript if published will not help my personal search for research funding. However, these authors provide a very convincing argument with ample evidence to support their conclusions.

Thank you.

Now I want to know what happens to the other 30 to 40% of northern Peatlands that are mineral poor to rich fens. The authors now clearly state that their results do not apply to fens. Our work suggests these may be much more sensitive to change. There is a group in Canada that is now attempting a similar experiment on fens and we need to convince some of these authors to bring their techniques a little further north.

We agree that investigation of fens would prove interesting.